

# PInc-PanTher estimates of Arctic permafrost soil carbon under the GeoMIP G6solar and G6sulfur experiments

Aobo Liu[1,2], Yating Chen[1,2], John C. Moore[2,3,4]

[1]College of Geography and Environment, Shandong Normal University, Jinan, 250014, China
[2]College of Global Change and Earth System Science, Beijing Normal University, Beijing, 100875, China
[3]CAS Center for Excellence in Tibetan Plateau Earth Sciences, Beijing, 100101, China
[4]Arctic Centre, University of Lapland, Rovaniemi, 96101, Finland

*Correspondence to*: Yating Chen (chenyt2016bnu@gmail.com), John C. Moore (john.moore.bnu@gmail.com)

**Abstract.** Circum-Arctic permafrost represents a tipping element for the Earth's climate system that must be maintained to avoid catastrophic climate change. Solar geoengineering (SG) has the potential to slow Arctic temperature rise by increasing planetary albedo, but could also reduce tundra productivity. Here, we improve the data-constrained PInc-PanTher model of permafrost carbon storage by including estimates of plant productivity and rhizosphere priming on soil carbon. Six earth system models are used to drive the model, running two SG schemes (G6solar and G6sulfur), and scenarios with substantive (SSP2-4.5) and no (SSP5-8.5) mitigation efforts. By 2100, simulations indicate that the permafrost area is expected to decrease by 9.2±0.4 (mean ± standard error), 5.6±0.4, 5.8±0.3, and 6.1±0.4 million km$^2$ and soil carbon loss will be 81±8, 47±6, 37±11, and 43±9 Pg under SSP5-8.5, SSP2-4.5, G6solar and G6sulfur, respectively. Uncertainties in permafrost soil C loss estimates arise mainly from changes in vegetation productivity due to climate warming and $CO_2$ fertilization. The increased input flux from vegetation to soil raises, while the priming effects of root exudates lowers soil C storage conservation, with the net effect mitigating soil C loss. Despite model differences, the protective effects of the G6solar and G6sulfur experiments on permafrost area and soil carbon storage are consistent and significant at the 95% level for all six ESM. SG mitigates ~1/3 of permafrost area loss and halves carbon loss for SSP5-8.5, averting about $20 trillion in economic losses by 2100 and might provide a sustainable income stream for the Arctic population.

**Short summary (500-character):**

Permafrost thaws and releases carbon (C) as the Arctic warms. Most Earth System Models (ESM) have poor estimates of C stored now and so their future C losses are much lower than using the PInc-PanTher permafrost C model with climate inputs from 6 ESM. Bias-corrected soil temperatures and vegetative productivity plus geoengineering lowering global temperatures from a "business as usual level to a moderate emissions level keep C in the soil worth about $20 trillion in climate damages by 2100.



## 1 Introduction

There is a roughly 50% chance that the world will limit warming to 2 °C above pre-industrial if current international carbon emission commitments are fully met (Liu et al., 2022; Meinshausen et al., 2022). Relying on increased carbon sinks or $CO_2$ removal to compensate for, or offset, fossil fuel emissions is not, at present, a reliable substitute for emissions controls (Fankhauser et al., 2022), and various effects of $CO_2$ already in the atmosphere will persist for centuries. Over recent decades the Arctic has warmed much faster than the global average (IPCC SROCC, 2020), placing much of the cryosphere, including the permafrost, at risk of thaw (Biskaborn et al., 2019). The large hysteresis in the water/ice phase change, and associated climate feedbacks make this an essentially irreversible change, that is a "tipping point" (Lenton et al., 2019). Permafrost covers 22% of the exposed land area in the Northern Hemisphere (Obu et al., 2019) and contains ~1000 Pg of soil organic carbon in its upper 3 m (Hugelius et al., 2014; Mishra et al., 2021). Deepening of the seasonally thawed permafrost active layer will induce microbial degradation of previously frozen soil organic carbon to $CO_2$ and $CH_4$, producing the permafrost carbon-climate feedback (PCF). An extreme warming scenario where no emissions mitigation occurs (RCP8.5 or similar) is expected to reduce the area of Arctic permafrost by 30–99% by 2100 and soil carbon storage by many tens of Pg C, with a multi-model average estimate of 92±17 Pg C (Schuur et al., 2015; IPCC SROCC, 2020).

Solar geoengineering (SG), a class of methods that limit or reverse anthropogenic climate change by reducing the amount of sunlight reaching Earth, has the potential to change Earth's climate, moderating climate hazards compared with pure greenhouse gas (GHG) scenarios (Irvine et al., 2019). The principal advantage of SG over $CO_2$ removal and substantial emission reductions is that temperatures can be reduced far faster; SG may also face fewer technical and financial hurdles (Aldy et al., 2021). In any case given that the potential damage caused by geoengineering has not yet been fully explored (Zarnetske et al., 2021), we recognize that the best option for mitigating climate change is to aggressively cut GHG emissions by switching to clean energy sources. It makes sense, however, to explore SG as a potential tool to avoid catastrophic climate change and is especially pertinent due to the rapidly growing profile of geoengineering within the scientific and policy community (Aldy et al., 2021; Keith, 2021). In an earlier study, Chen et al. (2020) found that five of seven CMIP5 Earth System Models (ESMs) driven by the G4 stratospheric aerosol injection geoengineering scheme simulated significant mitigation of Arctic permafrost soil carbon loss. The G4 scenario specifies an injection of $SO_2$ into the equatorial lower stratosphere at an annual rate equivalent to about 1/4 of the 1991 Mt. Pinatubo volcanic eruption under the RCP4.5 moderate GHG emissions scenario. The ESMs used and specification of the geoengineering scenarios have since been improved for the CMIP6 round of experiments.

We investigate the response of permafrost area and soil C stocks to GHG and SG imposed changes in radiative forcing, comparing two SG schemes (G6solar and G6sulfur), as well as high (SSP5-8.5) and medium (SSP2-4.5) GHG scenarios. The G6solar and G6sulfur experiments are part of the Geoengineering Model Intercomparison Project Phase 6 (GeoMIP6, Kravitz et al., 2015), which specifies a reduction in radiative forcing from the SSP5-8.5 to SSP2-4.5 levels by solar irradiance reduction or stratospheric aerosol injection respectively. G6 may represent more slightly more realistic simulations than previous





scenarios such as G4, since the amount of SG required to meet the goal varies over time and is dependent on ESM climate and sulfate aerosol sensitivity. The implementation of the G6sulfur experiment varies with the sophistication of the ESM atmospheric physical-chemistry component, ranging from calculations dependent on $SO_2$ emissions to prescribed aerosol optical depth distributions (Visioni et al., 2021). We use the outputs of the six CMIP6 ESMs (Table 1) to drive a data-constrained, process-based permafrost carbon model and compare the results it gives to the soil carbon changes calculated

directly within each of the ESMs. The primary objective is to determine the effects of implementing the G6solar and G6sulfur experiments on permafrost area and soil carbon loss, and to assess whether the six ESMs produce a response consistent with expectations that the cooling effect of SG will inhibit Arctic warming, slow permafrost degradation, and reduce PCF-induced GHG emissions. We then assess the economic benefits of G6-type geoengineering proposals for avoiding catastrophic permafrost degradation through an integrated assessment model that links the warming potential of the PCF to the

corresponding economic impacts.

**Table 1. The CMIP6 models used in this study and their main features.** The attributes of ESMs listed here are relevant to the soil temperatures (TSL), net primary productivity (NPP) and gross primary productivity (GPP) outputs, which are used to drive a mechanistic soil carbon model for simulating permafrost carbon dynamics. G6sulfur aerosols indicates use of prescribed aerosol optical depth (AOD) or internally calculated from injected $SO_2$ (Visioni et al., 2021).

| No. | ESM | Reference | G6sulfur aerosols | Land model | Resolution (lon×lat) | Maximum soil depth (layers) |
|---|---|---|---|---|---|---|
| 1 | CESM2-WACCM | Gettelman et al. (2019) | From $SO_2$ | CLM5 | 143 × 144 | 48.6 m (25) |
| 2 | CNRM-ESM2-1 | Séférian et al. (2019) | AOD | Surfex 8.0c | 192 × 288 | 12.0 m (14) |
| 3 | IPSL-CM6A-LR | Boucher et al. (2020) | From $SO_2$ | ORCHIDEE-2.0 | 128 × 256 | 90.0 m (18) |
| 4 | MPI-ESM1-2-HR | Mauritsen et al. (2019) | AOD | JSBACH3.2 | 192 × 384 | 9.8 m (5) |
| 5 | MPI-ESM1-2-LR | Mauritsen et al. (2019) | AOD | JSBACH3.2 | 96 × 192 | 9.8 m (5) |
| 6 | UKESM1-0-LL | Sellar et al. (2020) | From $SO_2$ | JULES-ES-1.0 | 144 × 192 | 3.0 m (4) |

| No. | ESM | Permafrost carbon | Dynamic vegetation | Nitrogen cycle | Snow density | Snow thermal conductivity |
|---|---|---|---|---|---|---|
| 1 | CESM2-WACCM | Yes | No | Yes | $f(T_{snow})$ | $f(\rho_{snow})$ |
| 2 | CNRM-ESM2-1 | No | No | No | $f(T_{snow})$ | $f(\rho_{snow})$ |
| 3 | IPSL-CM6A-LR | No | No | No | Fixed | Fixed |
| 4 | MPI-ESM1-2-HR | No | Yes | Yes | Fixed | Fixed |
| 5 | MPI-ESM1-2-LR | No | Yes | Yes | Fixed | Fixed |
| 6 | UKESM1-0-LL | No | Yes | Yes | $f(T_{snow})$ | $f(\rho_{snow})$ |



## 2 Materials and Methods

### 2.1 ESM carbon estimates

CMIP6 ESMs have many improvements over preceding generations of CMIP models, these include better treatment of snow radiative transfer and insulation effects, soil hydrology and vegetation dynamics, but estimates of permafrost extent and carbon stock changes are still very variable across models, and which has been associated with deficiencies in the representation of soil thermodynamics and carbon dynamics (Burke et al., 2020; Mudryk et al., 2020; Fox-Kemper et al., 2021). Of the six CMIP6 models we used (Table 1), only CESM2-WACCM was tuned for permafrost carbon stocks based on observations. The other five ESMs estimated initial Arctic permafrost carbon stocks (CSoil) between 67–475 Pg (Table 2), which are much smaller than observed estimates (Hugelius et al., 2014; Mishra et al., 2021). MPI-ESM1-2-HR does not report CSoil but it shares many features with MPI-ESM1-2-LR, and so may be expected to have similar Csoil. CESM2-WACCM simulates a slight decrease in CSoil under SSP5-8.5, while the other ESMs show an increase in CSoil (Table 2). This can be explained by the underestimation of initial permafrost CSoil which then leads to an underestimation of future soil C loss. CSoil flux from surface vegetation increases due to GHG-driven increases in productivity, and this flux depends on permafrost area which dominates CSoil losses in those models with little initial soil carbon.

Maximum soil depth, number of soil layers, and snow schemes are important for soil temperatures (TSL) simulation (Wang et al., 2016). CESM2-WACCM, CNRM-ESM2-1 and IPSL-CM6A-LR simulate more than 10 soil layers with a maximum depth greater than 10 m (Table 1), potentially producing more accurate soil temperature profiles compared with models that have shallower soil depths and fewer layers. Parametric schemes that model snow density and snow thermal conductivity as a function of snow temperature and snow density (Table 1) and take into account the mechanical compaction process of snow, may provide more accurate TSL simulations than those using fixed snow parameters (Wang et al., 2016). In addition, ESM with dynamic vegetation and nitrogen cycle (MPI-ESM1-2-HR, MPI-ESM1-2-LR and UKESM1-0-LL) may be able to more accurately model changes in vegetation C pools and input fluxes to soil C pools (Table 1).





**Table 2. CMIP6 ESM initial permafrost soil and vegetation C stocks and their changes.** Values are calculated based on
the CSoil and CVeg outputs (in Pg) of five CMIP6 ESMs, and are clipped to the extent of the circum-Arctic continuous
permafrost zone. For the changes in CSoil and CVeg between 2020 and 2100, '-' indicates net C loss and '+' indicates net C
gain. MPI-ESM1-2-HR does not report CSoil but is likely similar as MPI-ESM1-2-LR.

| initial C stocks | CESM2-WACCM | CNRM-ESM2-1 | IPSL-CM6A-LR | MPI-ESM1-2-LR | UKESM1-0-LL |
|---|---|---|---|---|---|
| CSoil in 2020 | 1058 | 475 | 67 | 156 | 274 |
| CVeg in 2020 | 68 | 65 | 24 | 32 | 53 |
| **CSoil change (2020–2100)** | | | | | |
| SSP5-8.5 | -20 | +36 | +15 | +9 | +21 |
| SSP2-4.5 | -6 | +32 | +17 | +9 | +21 |
| G6solar | -4 | +48 | +17 | +12 | +26 |
| G6sulfur | -3 | +45 | +16 | +11 | +25 |
| **CVeg change (2020–2100)** | | | | | |
| SSP5-8.5 | +47 | +64 | +59 | +16 | +43 |
| SSP2-4.5 | +27 | +38 | +43 | +13 | +33 |
| G6solar | +34 | +59 | +47 | +15 | +36 |
| G6sulfur | +33 | +57 | +40 | +14 | +36 |

## 2.2 Bias correction for ESM simulations

Since the ESM produces great differences in the key parameters that drive carbon storage in the permafrost region, we eliminate
the systematic errors in ESM simulations by a bias correction procedure. We bias-correct the TSL, NPP, and GPP outputs of
the ESMs with the trend-preserving, Inter-Sectoral Impact Model Intercomparison Project (ISI-MIP) method (Hempel et al.,
2013). This method produces a distribution of each parameter that matches the mean of the reference field over its record
length, but maintains the ESM-dependent time trend of each field. The bias correction process downscales the data to match
the chosen reference field, in this case the 0.1° × 0.1° reanalysis data and satellite observations from 2015 to 2020. Our study
region excludes sporadic/isolated-patches of permafrost and dynamics of non-permafrost soil orders occurring within
discontinuous permafrost regions, because we are primarily concerned with forced, large-scale permafrost degradation and
mitigated soil carbon loss in response to the cooling effects of SG.

The TSL outputs of ESMs are bias-corrected using soil temperatures from ERA5-Land reanalysis data (Muñoz-Sabater et al.,
2021). Figure 1 shows the need for initial bias correction before assessing permafrost degradation and soil C loss, since the
soil and snow physical processes that determine the thermal properties of permafrost soils differ greatly among the CMIP6
ESMs. Before bias correction, the mean annual soil temperatures simulated by four ESM are 2 to 4 °C lower than that of the
ERA5-Land. Since decomposition rate is a nonlinear function of TSL, lower simulated TSL leads to an underestimation of
soil C loss. The upward trends in soil temperature are preserved after bias correction, and the G6solar and G6sulfur experiments





effectively reduce soil temperature to levels similar to those of the SSP2-4.5 simulations (Fig. 1), despite GHG levels being as
in SSP5-8.5. The six CMIP6 ESMs simulated soil temperatures in the circum-Arctic permafrost region increase by 7.2±0.2
(mean±s.e.), 3.3±0.2, 3.5±0.2, and 3.6±0.3 °C between 2020 and 2100 under the SSP5-8.5, SSP2-4.5, G6solar, and G6sulfur
scenarios, respectively.

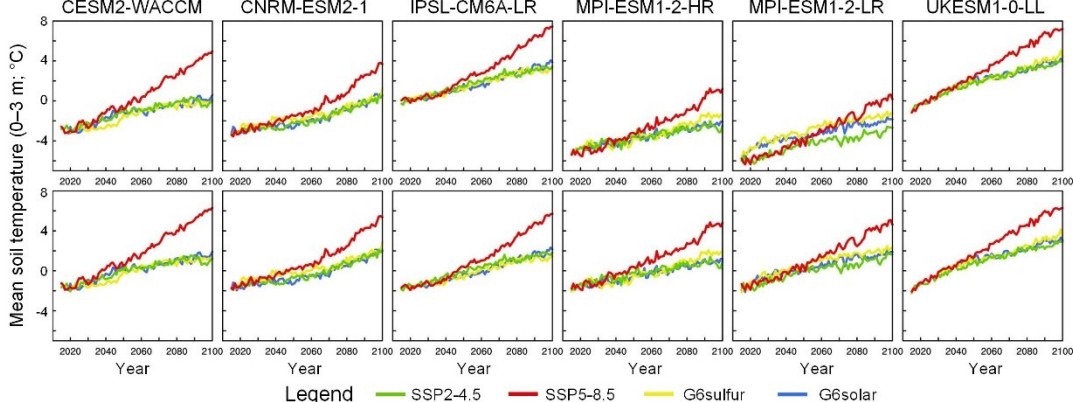

**Figure 1.** Bias-correction of soil temperature (TSL). Mean annual soil temperature in the upper 3 m of the circum-Arctic
permafrost region before (top row) and after (bottom row) bias-correction.

The NPP and GPP outputs of ESMs are bias-corrected using the corresponding MODIS products (Running et al., 2019).
Simulations show that both vegetation productivity and root activity in the permafrost region are expected to increase in the
future (Figs. 2 and 3), due to rising temperature and atmospheric $CO_2$ fertilization. The increase in plant productivity will lead
to greater input fluxes from plant litter to the soil C pool, but on the other hand, plant root exudates will alter soil pH values
and enhance microbial decomposition activity through the rhizosphere priming effect (RPE). The RPE ratio is defined as the
ratio of respiration from plant-affected over non-plant-affected soils, which can be estimated from GPP based on the relation
found by a wide-ranging meta-analysis of permafrost soils (Keuper et al., 2020). The NPP and RPE simulations under the
G6solar and G6sulfur experiments are expected to fall between the SSP5-8.5 and SSP2-4.5 scenarios, because the SG reduces
the temperature but not the atmospheric $CO_2$ concentration. CNRM-ESM2-1 simulates the greatest rises in NPP and RPE by
2100, while IPSL-CM6A-LR is notably lower under the G6sulfur scenario than the other ESM, and the remaining four models
simulate similar trends (Figs. 2 and 3).





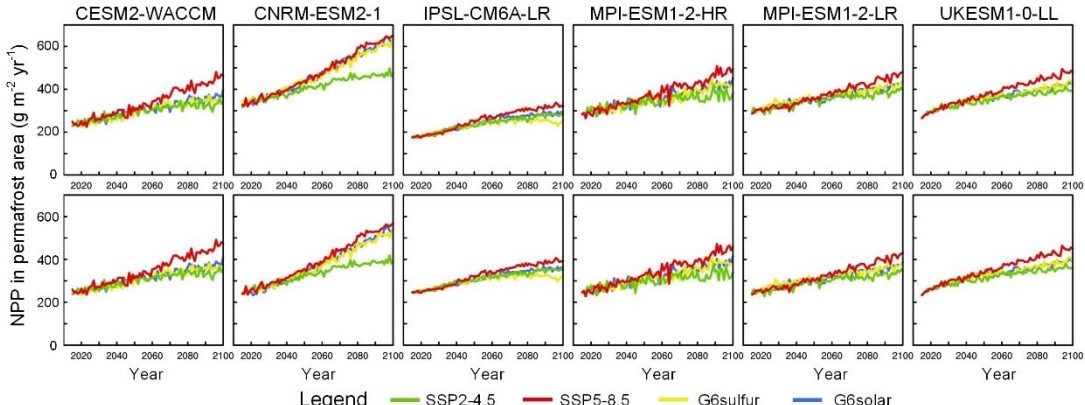

**Figure 2:** Bias-correction of net primary productivity (NPP). Mean NPP in the permafrost region before (top row) and after (bottom row) bias-correction.

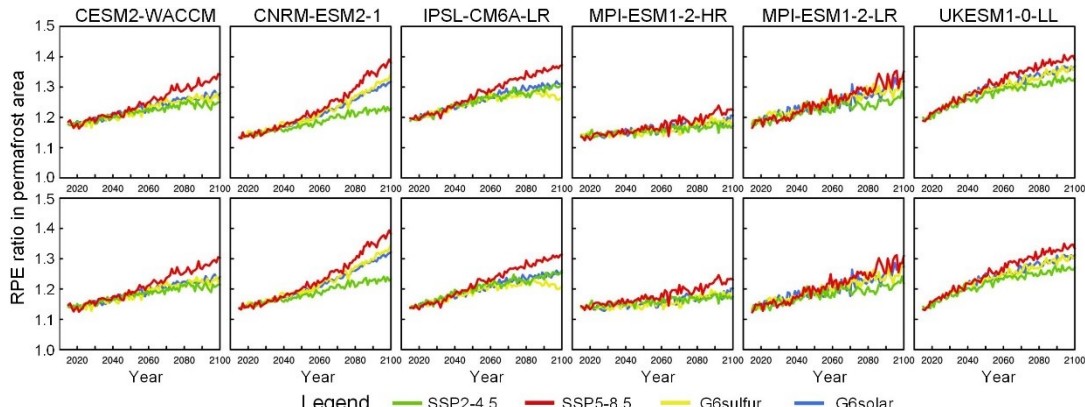


**Figure 3:** Bias-correction of rhizosphere priming effect (RPE) ratio. Mean RPE ratio in the permafrost region before (top row) and after (bottom row) bias-correction. RPE ratio is calculated from gross primary productivity (GPP).

## 2.3 The modified PInc-PanTher model

Instead of using the carbon stock and carbon flux outputs of ESMs for our primary analysis, we use bias-corrected ESM output

of TSL, NPP and RPE to drive a specially modified PInc-PanTher model to estimate permafrost carbon. PInc-PanTher, a data-constrained model developed by Koven et al. (2015) for estimating PCF, is characterized by compiled soil C maps and laboratory incubation syntheses specifically from permafrost soils. Permafrost soil C maps are derived from the thematic soil classification maps published by the Permafrost Carbon Network (Hugelius et al., 2014), with a horizontal resolution of 0.1° and divided vertically into four layers: 0–0.3, 0.3–1, 1–2, and 2–3 m. The initial permafrost soil C storage in the upper 3 m of

soils is 727 Pg C, which excludes non-permafrost soils with 280 Pg C in the discontinuous permafrost regions (Hugelius et al., 2014). While typical terrestrial C models use a single set of decomposition constants to model the transient dynamics of soil C losses, PInc-PanTher builds a parallel three-pool, first-order decomposition model based on soil incubation meta-analysis to calculate a set of parameters that best describe permafrost C losses for the laboratory incubations (Koven et al., 2015). The



fraction of each pool in the decomposition constants can be determined in two ways: dependent on the soil C:N ratio, and
dependent on the ratio of mineral and organic matter in the soil (Schädel et al., 2014); in this study we use the average of these
two methods.

The standard PInc-PanTher model considers only the effect of TSL on microbial decomposition and ignores vegetation
dynamics, i.e., the input C flux into the soil pool is fixed and the effect of plant roots is not considered. We introduce time-
varying input C fluxes and RPE ratios into the modified PInc-PanTher model to reflect the effects of plant productivity changes
on microbial decomposition and permafrost soil C storage. The change in permafrost C stock (Cp) with time can be expressed
as a function of NPP, TSL and RPE as:

$$\frac{\mathrm{d}C_p}{\mathrm{d}t} = P(NPP, t) - C_p \cdot k \cdot Q_{10}(\mathrm{TSL}) \cdot \mathrm{RPE}_{ratio}, \quad (1)$$

where $P(NPP, t)$ is the time-dependent input C flux, which we assume to be proportional to NPP on annual scales in every soil
layer. The initial carbon flux into the soil pool is inferred from the initial steady state, which satisfies the condition that soil C
loss and input are in equilibrium during 2015–2019. $k$ is the decay constant at the reference temperature (5 °C) which equals
the inverse of the turnover time. $Q_{10}(TSL)$ is a function of soil temperature that controls the decomposition rates, and which
assumes a 2.5-fold increase in respiration rate for each 10 °C increase in soil temperature. Given the strong nonlinear
relationship between decomposition rate and soil temperature, PInc-PanTher calculates decomposition rate is based on the
temperature of each month and then averaged over the course of a year rather than using the annual mean temperature directly
for the calculation. The RPE ratio reflects the effect of plant roots on soil respiration rate and is calculated as:

$$\mathrm{RPE}_{ratio} = 1 + \frac{2.47 \times Respiration_{Root}}{13.01 + Respiration_{Root}}, \quad (2)$$

where root respiration is estimated to be 3.6% of GPP (Keuper et al., 2020). RPE-induced respiration is dominated by the
shallow soil layer because most roots are the upper soil; thus we fix the RPE ratio to 1 for soil layers >0.3 m in depth.

## 2.4 PAGE-ICE integrated assessment model

PAGE-ICE is an updated version of the PAGE (Policy Analysis of Greenhouse Effects) model (Hope and Schaefer, 2016),
ICE stands for Ice, Climate and Economy, and incorporates the permafrost carbon feedback into the economic impact
assessment of climate change (Yumashev et al., 2019). PAGE-ICE includes multiple updates to climate science and economics
from IPCC AR5 and subsequent literature, providing economic assessments of climate-driven impacts in four sectors (Liu et
al., 2022), which include economic (damages to overall economy), non-economic (public health and ecosystem services), sea
level rise (relocation and coastal flood damage), and discontinuity (large-scale damage associated with climatic tipping points).
Damages to infrastructure from permafrost degradation are not included in the PAGE-ICE economic impact assessment, but
have a non-negligible impact (Hjort et al., 2022).

Here, we use PAGE-ICE to assess the socioeconomic benefits of G6 experiments for mitigating permafrost degradation and
maintaining the stability of permafrost C storage. The G6 experiments are assumed to follow the same socioeconomic





trajectory and emission pathway as the baseline scenario SSP5-8.5. We first simulate the impact of the G6 experiments to reduce permafrost $CO_2$ and $CH_4$ emissions on future climate change, based on the simplified climate module built into PAGE-ICE. We then use the damage function "PAGE09 & IPCC AR5 & Burke" provided by PAGE-ICE, which is chosen to make the assessment comparable with existing studies (Hope and Schaefer, 2016; Chen et al., 2020), linking global and regional warming to GDP losses. Finally, the estimated climate damages are processed using the "equity-weighted ON, PTP

discounting" scheme provided by the discounting module of PAGE-ICE, converting changes in consumption to utility to correct for regional income differences, and discounting aggregates based on the pure time preference rate.

## 3 Results

### 3.1 Projected permafrost area loss

Warming causes widespread degradation of permafrost, and Figure 4 shows the changes in circum-Arctic permafrost area over

time for the period 2020–2100 as diagnosed by a mean annual ground temperature (MAGT) of 0 °C at 3 m depth, which is approximately the bottom of the active layer. Simulations of permafrost area change across ESMs show a high degree of agreement (Fig. 4), except for the UKESM1-0-LL which shows greater area reduction due to it having the largest simulated warming (Fig. 1). Permafrost area, and other results we give are as the multi-model mean ± standard error, are projected to decrease by 9.2±0.4, 5.6±0.4, 5.8±0.3, and 6.1±0.4 million km² under the SSP5-8.5, SSP2-4.5, G6solar, and G6sulfur scenarios,

respectively. Simulations by McGuire et al. (2018) suggested a loss of permafrost area of 4.1±0.6 and 12.7±5.1 million km² by 2300 for the RCP4.5 and RCP8.5 scenarios, respectively. Here we expect faster and more extensive permafrost degradation because the CMIP6 ESMs we use simulate faster warming rates than those used in McGuire et al. (2018).

The soil temperature cooling under the G6 experiments and the slowing permafrost degradation is significant (Table 3). By 2100, compared with the SSP5-8.5 scenario, the G6solar and G6sulfur experiments will reduce annual mean soil temperatures

in the permafrost region by 3.5±0.2 and 3.3±0.4 °C, respectively, and mitigate 3.4±0.2 and 3.1±0.5 million km² of permafrost degradation. G6sulfur produces a weaker response with larger uncertainty than G6solar, as may be expected given the ESM-dependent implementation of the radiative response to aerosols in the G6sulfur scheme compared with the much simpler G6solar simulation. Despite model differences, the protective effects of the two G6 schemes on permafrost extent are consistent and significant at the 95% level for all six ESMs (Table 3).

Figure 5 shows the spatial distribution of the simulated permafrost extent in 2100 for the four scenarios. Only small patches of permafrost in the Canadian Archipelago and central Siberia are preserved under the SSP5-8.5 scenario, while more permafrost at high latitudes remains under the SSP2-4.5 and G6 scenarios (Fig. 5). Large-scale degradation of circum-Arctic permafrost is simulated even under moderate mitigation and G6 geoengineering scenarios. Relative to the SSP5-8.5 scenario, the G6 experiments can reduce permafrost area loss by about 1/3 on average, i.e., protect about 3 million km² of permafrost

area (Table 3).



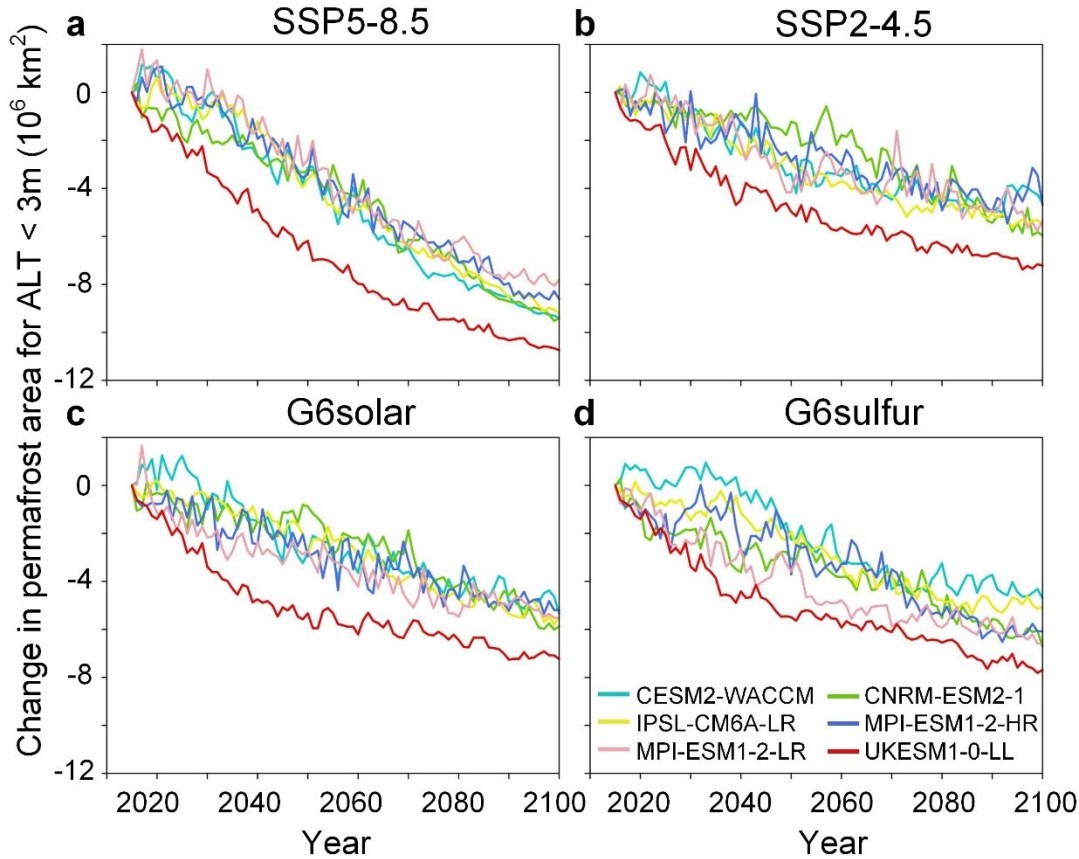

**Figure 4:** Cumulative change in circum-Arctic permafrost area derived from bias-corrected TSL. Simulations are performed under the SSP5-8.5, SSP2-4.5, G6solar and G6sulfur scenarios up to year 2100.





**Table 3. G6 impacts on soil temperatures, permafrost degradation, and soil C storage.** Cooled soil temperature (in °C), preserved permafrost area (in million km$^2$), and retained permafrost soil C (in Pg) at the end of the century for the G6solar and G6sulfur scenarios relative to the SSP5-8.5 scenario. Bold fonts, that is all values, are significant at 95 % level according to the Wilcoxon signed-rank test.

| Cooled TSL (°C) | CESM2-WACCM | CNRM-ESM2-1 | IPSL-CM6A-LR | MPI-ESM1-2-LR | MPI-ESM1-2-LR | UKESM1-0-LL | Ensemble |
|---|---|---|---|---|---|---|---|
| G6solar | **4.4** | **3.4** | **3.6** | **3.6** | **2.7** | **3.2** | **3.5** |
| G6sulfur | **4.9** | **2.5** | **4.3** | **2.9** | **2.3** | **2.8** | **3.3** |
| **Preserved area (million km$^2$)** | | | | | | | |
| G6solar | **4.1** | **3.5** | **3.7** | **3.4** | **2.3** | **3.5** | **3.4** |
| G6sulfur | **4.7** | **2.7** | **4.1** | **2.5** | **1.2** | **3.0** | **3.1** |
| **Retained Soil C (Pg)** | | | | | | | |
| G6solar | **49** | **61** | **43** | **47** | **32** | **38** | **45** |
| G6sulfur | **49** | **44** | **45** | **33** | **25** | **32** | **38** |


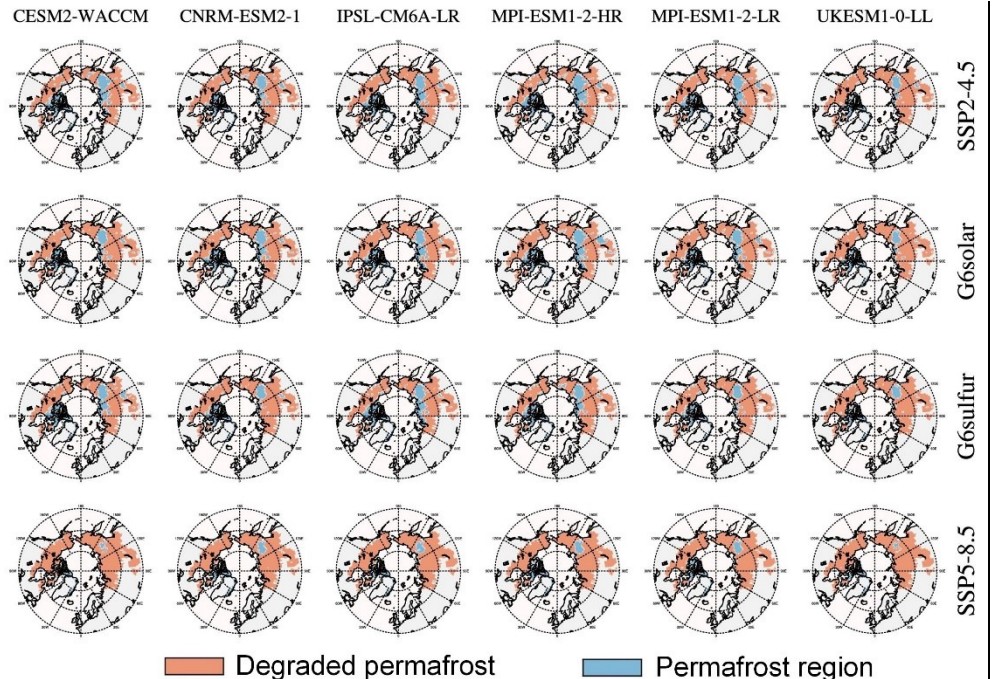

**Figure 5:** Maps of permafrost extent in the year 2100, derived from the bias-corrected TSL outputs of six ESMs. Permafrost regions with active layer depths greater than 3 meters in 2100 are considered degraded permafrost.





## 3.2 Projected permafrost C loss

Based on the modified PInc-PanTher model and bias-corrected TSL, NPP and RPE simulations, we calculate the change in permafrost soil C over time for the period 2020–2100 (Fig. 6). Permafrost soil C is expected to lose 81±8, 47±6, 37±11, and 43±9 Pg C under the SSP5-8.5, SSP2-4.5, G6solar and G6sulfur scenarios, respectively. Our estimated permafrost C losses for the SSP5-8.5 projection are close to the 92±17 Pg C estimate based on a literature compilation (Schuur et al., 2015). The permafrost C losses under the SSP2-4.5 scenario we simulate are within the central part of the range of 11–135 Pg C reported

by Burke et al (2013), and higher than the range of 12.2–33.4 Pg C reported by Koven et al (2015). There is considerable variation in the predictions of permafrost C loss among the models (Fig. 6), with UKESM1-0-LL simulating the most, and CNRM-ESM2-1 the least carbon losses under all scenarios. This follows from UKESM1-0-LL predicting the most pronounced TSL rises, while CNRM-ESM2-1 predicts the greatest NPP increases (Figs. 1 and 2).

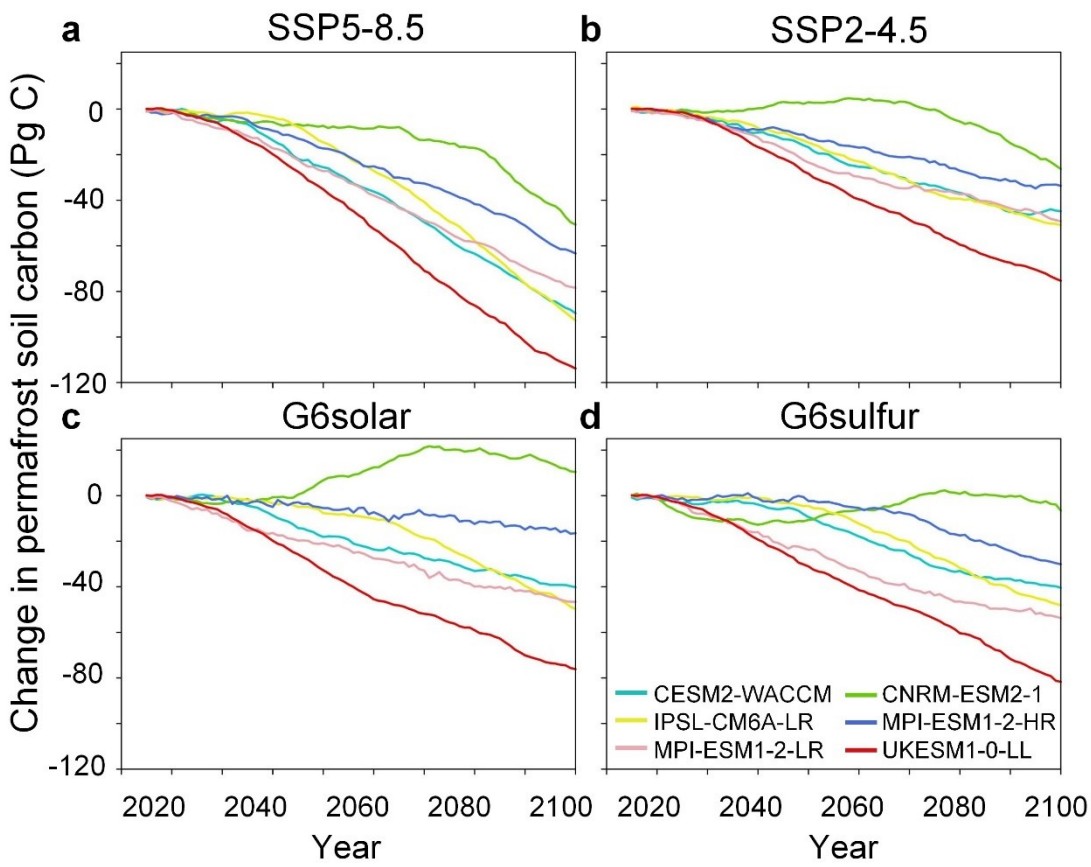


**Figure 6:** Cumulative change in circum-Arctic permafrost C storage obtained from modified PInc-PanTher driven by TSL, NPP and RPE simulations. Simulations are performed under the SSP5-8.5, SSP2-4.5, G6solar and G6sulfur scenarios up to year 2100.





There are model differences and spatial heterogeneity in the simulated soil C loss (Fig. 7), and the spatial distribution is
controlled by latitude and initial C stocks. Core permafrost regions such as Siberia play a dominant role, while the southern
permafrost margin is less affected (Fig. 7). The reason for this is that permafrost soils at the southern edges are usually already
in a state of seasonal or permanent thaw and remain so as the climate warms, in contrast, at high Arctic latitudes where much
carbon has accumulated, rising summer temperatures greatly prolong the thawing time and promote decomposition.

Compared with SSP5-8.5, G6solar and G6sulfur retain 45±4 and 38±3 Pg C (Table 3), respectively, almost halving the
permafrost soil C loss. G6solar and G6sulfur provide more protection against permafrost soil C loss than does SSP2-4.5,
mainly due to higher vegetation productivity (Fig. 2) driven by their higher GHG concentrations. In addition, soil C losses for
G6solar and G6sulfur differ from each other in some regions such as North America and European Russia more than in other
regions. This may be related to changes in tropospheric dynamic circulation caused by stratospheric sulfate aerosols (Visioni
et al., 2020), although little research has been done on such large scale circulation changes to date.

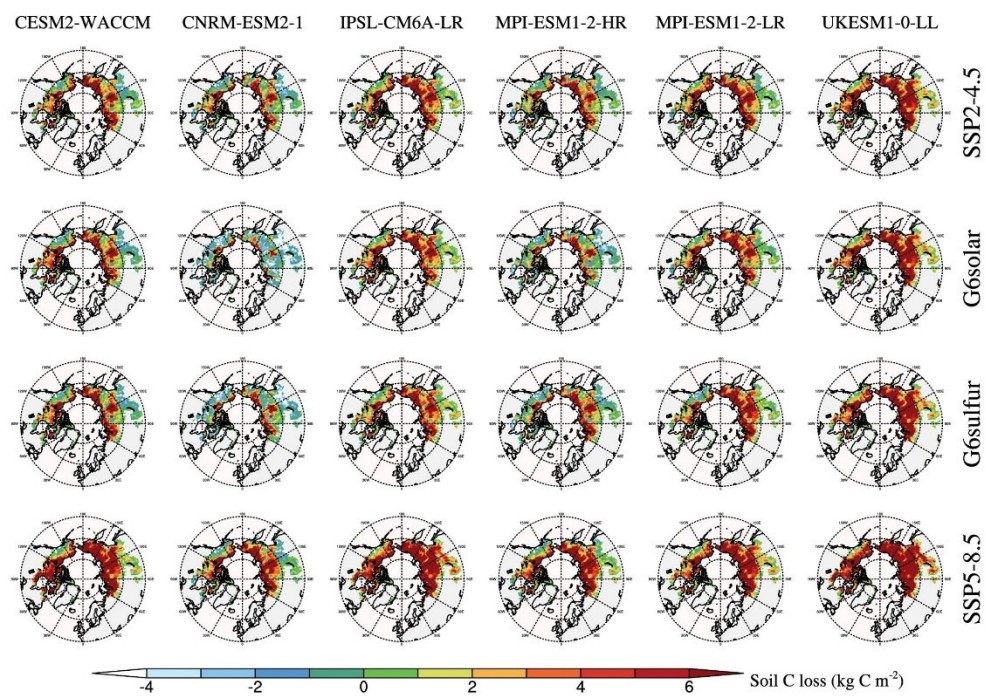


**Figure 7:** Maps of permafrost soil C losses between 2020 and 2100 integrated from surface to 3 m depth, obtained from
modified PInc-PanTher driven by TSL, NPP and RPE simulations.

### 3.3 Sources of uncertainty in C estimates

How differences in TSL, RPE, and NPP affect permafrost soil C is shown in Fig. 8. The original PInc-PanTher model only
considers the effect of temperature on soil respiration, hence the soil C loss is dominated by TSL, and the C losses simulated
under the SSP2-4.5, G6solar and G6sulfur scenarios are very similar, with relatively small uncertainties. Changes in vegetation
productivity due to warming and $CO_2$ fertilization have both positive and negative effects on the permafrost carbon-climate



feedback (Koven et al., 2015). On the one hand, shrub expansion in the tundra and poleward displacement of the tundra-taiga ecotone boundary will change the distribution of plant productivity and increase C inputs to the soil. On the other hand, plant

root exudates have priming effects on soil C turnover and will speed up the decomposition of organic matter. The positive and negative effects of vegetation partially offset each other, with the overall effect being to mitigate soil C loss.

Soil C losses under the two G6 scenarios are slightly lower than that of the SSP2-4.5 scenario after taking the effect of vegetation into account, but with clearly larger uncertainties for G6solar than the other scenarios. Specifically, RPE will increase the net loss of soil C, causing additional permafrost emissions of about 6 to 10 Pg C between 2020 and 2100. Our

estimates are within, but towards the lower range of those reported by Keuper et al (2020), who estimated RPE-induced soil C loss to be 5.9–75 (mean 38) Pg C for RCP 4.5 and 6.0–80 (mean 40) Pg C for RCP 8.5. PInc-PanTher simulates the steady-state equilibrium of the input C flux and decomposition rate during initialization, whereas RPE ratios rise from about 5% to 15% between 2020 and 2100, leading to additional soil C losses (Fig. 3). In comparison, the change in NPP is more dramatic, increasing by between ~40% to ~80% from 2020 to 2100 (Fig. 2). Increasing input C flux allows more carbon to be retained

in the permafrost soils, thus the lower C loss and larger uncertainty under the G6solar and G6sulfur scenarios compared with SSP2-4.5 is mainly caused by the large across model spread in NPP change under the changed direct and diffuse radiative forcing (Fig. 8).

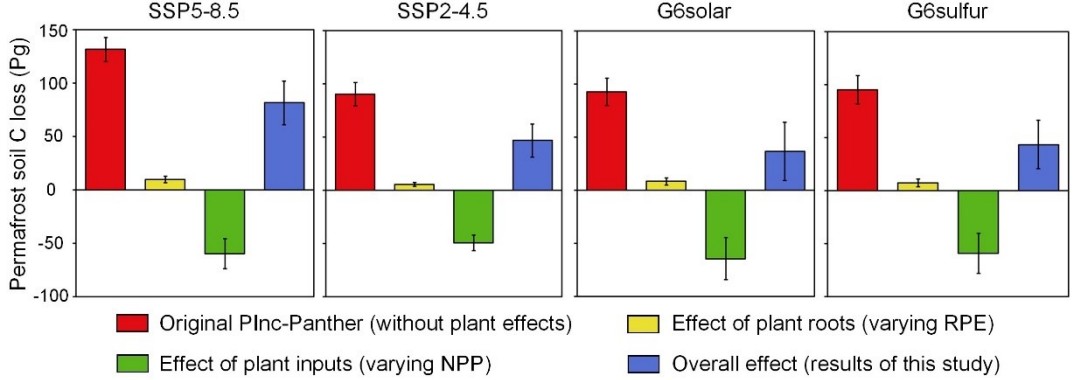

**Figure 8:** Cumulative permafrost soil C loss between 2020 and 2100 obtained from the original PInc-PanTher model (without
plant effects) and the modified PInc-PanTher model we used (with varying NPP and RPE), as well as the effects of plant inputs (NPP) and roots (RPE) on soil C loss. Bars represent the ensemble mean of the six ESMs, with the error line range being the standard deviation. Negative values represent permafrost C gain.

We further analyzed the differences among ESMs and find that the three models with better snow schemes (CESM2-WACCM, CNRM-ESM2-1 and UKESM1-0-LL) exhibit seasonal differences in warming, while the other three models with fixed snow

parameters (Table 1) simulate little or no seasonal differences in warming (Fig. 9). The exponential relationship between soil temperature and respiration rate suggests that a 1 °C increase in soil temperature has a greater effect on permafrost soil C decomposition if it occurs in warmer months than cold months. Among all ESMs, UKESM1-0-LL produces the most prominent warming in July-September, with SSP5-8.5 producing a warming of ~12 °C, and the other three scenarios producing warmings of ~8 °C, thus simulating the largest soil C loss. Under the SSP5-8.5 scenario, CESM2-WACCM and CNRM-





ESM2-1 have a similar mean annual warming of about 8 °C as does UKESM1-0-LL, but with a warm season 4 °C cooler (Fig.
9), so their simulated C losses are lower than UKESM1-0-LL (Fig. 6). MPI-ESM1-2-HR and MPI-ESM1-2-LR share the same
land surface module at different resolutions (Table 1), but produce TSL, NPP and RPE simulations that differ both in range
and spatial distribution, resulting in C loss estimates that can vary by 15–30 Pg (Fig. 6). Overall, parametric differences among
ESMs and uncertainties associated with future simulations cannot overshadow the strong indications that G6 solar and sulfate

geoengineering can effectively suppress Arctic warming, slow permafrost degradation, and conserve soil C storage (Table 3).

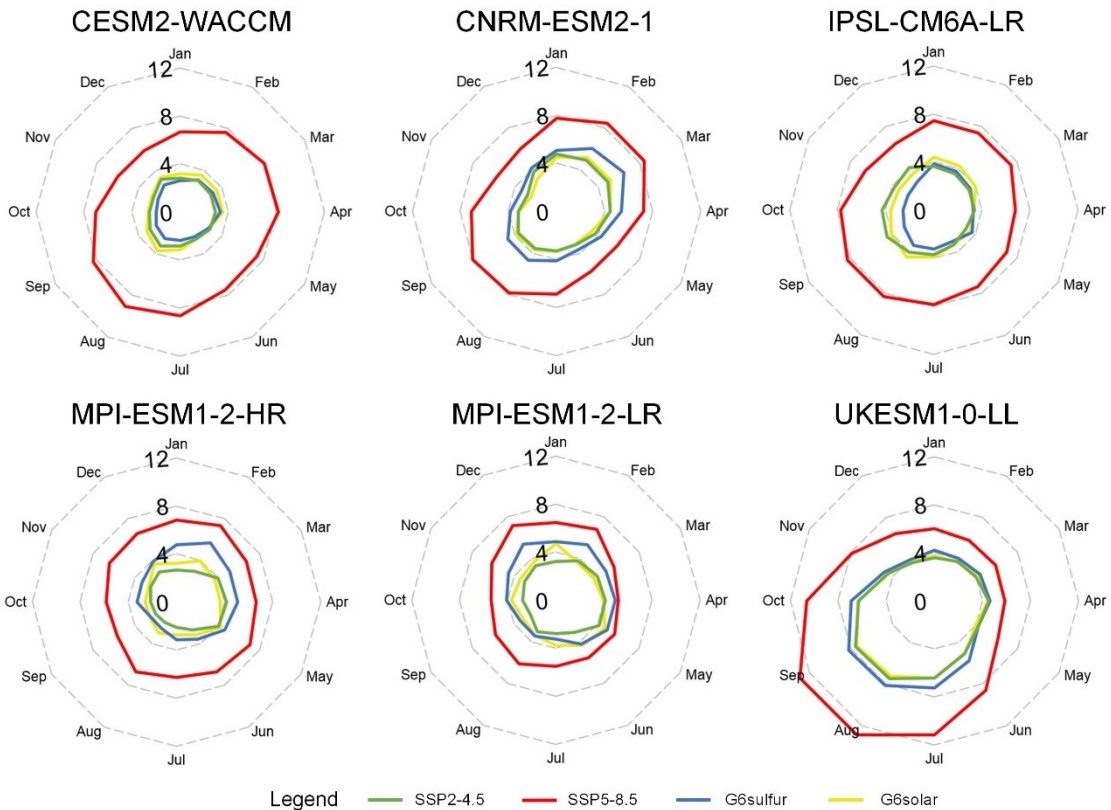

**Figure 9:** Monthly soil temperature change between 2020 and 2100. Warming in monthly soil temperature (units: °C) for each
ESM over the period 2020-2100 under the SSP2-4.5, SSP5-8.5, G6sulfur and G6solar scenarios.

### 3.4 Economic benefits from retained permafrost C

We estimate the economic benefits of permafrost soil C retained under the G6 simulations using the latest PAGE-ICE
integrated assessment model (Yumashev et al., 2019). PAGE-ICE has been widely used in the assessments of climate policy
and the social cost of carbon and, of high relevance to this study, the additional costs associated with Arctic permafrost
degradation (Yumashev et al., 2019; Chen et al., 2020; Liu et al., 2022). The overall economic impacts of SG are highly
complex and include both positive benefits of avoided warming and a variety of potential negative impacts, with many

unknown unknowns (Irvine et al., 2019; Zarnetske et al., 2021). Here we evaluate, based on PAGE-ICE, the economic impacts





of the reduced $CO_2$ and $CH_4$ emissions by SG through attenuated PCF, which is the less controversial part of the economic impact assessment related to geoengineering (Chen et al., 2020).

Figure 10 illustrates the reduction in permafrost $CO_2$ and $CH_4$ emissions due to the cooling effect of SG for the G6solar and G6sulfur scenarios compared with the baseline SSP5-8.5 scenario, and their cumulative economic impacts over the period
2020–2100. The permafrost $CO_2$ and $CH_4$ emissions reduced by the two G6 schemes do not show significant differences from each other, and their values show an approximately linear increasing trend with time, with a wide range of uncertainty. By 2100, the reduced permafrost emissions are about 4 Gt $CO_2$ and 16 Mt $CH_4$ per year (Fig. 10). Methane emissions from permafrost are influenced by many factors such as the degree of waterlogging, soil carbon-nitrogen ratio and local biome (Treat et al., 2015). We assume that $CH_4$ emissions are 2.3% of the overall soil respiration rate based on available studies and
expert assessments (Schuur et al., 2013; Schneider Von Deimling et al., 2015; Gasser et al., 2018; Chen et al., 2020), and that the remaining soil carbon is emitted as $CO_2$.

Assuming the same socioeconomic trajectory as SSP5-8.5, we simulate the economic benefits of the reduced permafrost emissions from SG (mean ± 1σ, as Gaussian distributions) by incorporating them into the PAGE-ICE IAM. We then estimate the cumulative sum of the discounted economic impacts over the period 2020–2100, considering the time value of money and
the loss of utility (Fig. 10c). All reported results are derived from an ensemble of 100,000 Monte Carlo simulations to perturb various model parameters, fully explore uncertainties in the economic impacts of climate damages, and establish probability distributions of the results. The simulations show that the 90% confidence intervals of the economic benefits due to the attenuated PCF are expected to be $0–70 (mean 20) and $0–67 (mean 18) trillion for G6solar and G6sulfur, respectively.

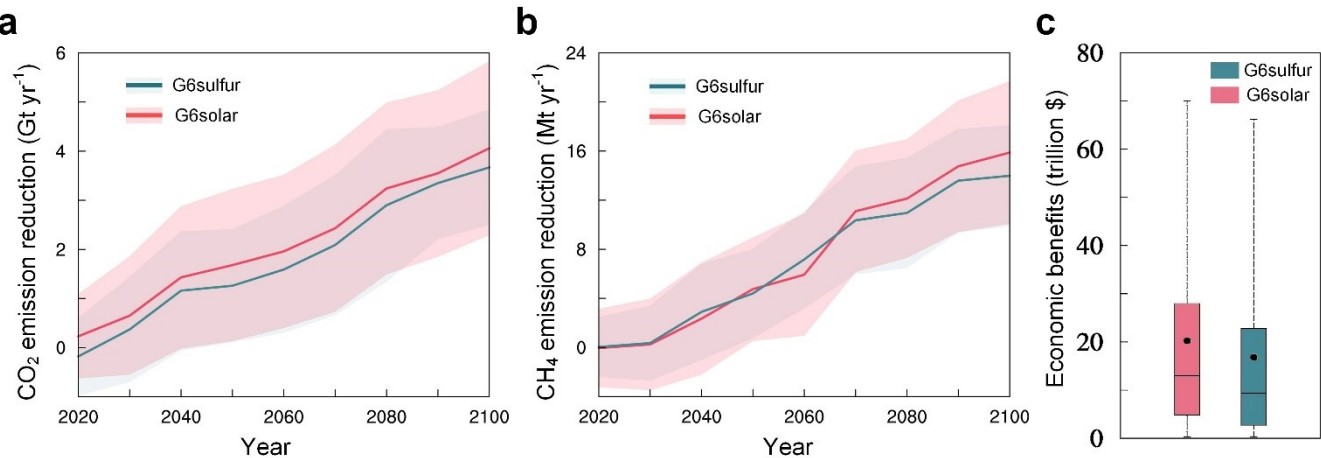

**Figure 10:** Reduced emissions of (a) $CO_2$ and (b) $CH_4$ due to retained permafrost C for G6solar and G6sulfur compared with the SSP5-8.5 scenario. Solid lines represent the mean of the six ESMs, and shaded areas represent the mean±standard deviation. (c) Cumulative economic benefits between 2020 and 2100 due to attenuated PCF, obtained from 100,000 Monte-Carlo runs of PAGE-ICE. Whiskers: 5–95% range; boxes: 25–75% range; horizontal lines: median; dots: mean.



## 4 Discussion and Conclusions

The modified PInc-PanTher model used in this study considers the effects of varying plant productivity on input fluxes and root activity and provides data-constrained estimates of the large-scale permafrost C response to warming. Circum-Arctic permafrost soil C is projected to lose 81±8, 47±6, 37±11, and 43±9 Pg C during 2020–2100 under SSP5-8.5, SSP2-4.5, G6solar, and G6sulfur scenarios, respectively. The permafrost soil C losses estimated in this study are close to estimates based on a literature compilation (Schuur et al., 2015), but differed in sign from the soil C outputs of most CMIP6 ESMs (Table 2). All

ESMs except CESM2-WACCM estimate permafrost soils to be a carbon sink over the 21st century rather than a carbon source (Table 2) because their simulated initial soil carbon stocks are much smaller than the observed estimates, leading to an underestimation of soil C loss. The bias correction process for TSL also shows that most ESMs underestimate soil temperatures in recent decades compared with the ERA5-Land reanalysis data (Fig. 1).

Gains in vegetation C are expected to partially, or even over-compensate, for losses in soil C in the permafrost region (McGuire

et al., 2018). The vegetation C pool (CVeg) in the permafrost region is expected to increase by 46±7, 31±5, 38±7 and 36±6 Pg C between 2020 and 2100 under the SSP5-8.5, SSP2-4.5, G6solar and G6sulfur scenarios, respectively (Table 2). Thus, for the circum-Arctic permafrost ecosystem, gains in vegetation C will largely offset losses in soil C. The higher plant productivity under the G6solar and G6sulfur experiments is expected to retain more C in the ecosystem compared with SSP2-4.5.

This study explores PCF estimates associated with the gradual thawing of permafrost as a response to deepening of the active

layer, loss of permafrost, soil warming, and the lengthening of the seasonal thaw period. Some important disturbances, such as tundra fires, thermokarst and thermoerosion, may lead to abrupt permafrost thaw and enhanced $CO_2$ and $CH_4$ emissions (Walter Anthony et al., 2018; Turetsky et al., 2020; Miner et al., 2022), are not included in the simulations due to both their limited implementation and physical understanding in process-based carbon models. However, the cooling effect of SG will certainly have a positive impact on avoiding abrupt permafrost thaw.

Our results show that for all six ESM simulations, the protective effect of the G6solar and G6sulfur experiments on permafrost area and soil C is significant at the 95% level (Table 3). Implementation of SG on the basis of the SSP5-8.5 scenario could reduce permafrost area loss by 1/3 and halve soil C loss, resulting in economic benefits of about $20 trillion by 2100. The implementation cost of the G6sulfur scheme is strongly dependent on scenario but for high greenhouse gas emission and consequent degrees of cooling is estimated at $30-70 billion/year (Smith, 2020), thus our experiments show that SG has

considerable global economic benefits. Whether there would be political and ethical benefits for the Arctic populations is a matter that those communities should consider with some urgency. Given the undoubted global benefits of preserving the Arctic permafrost, there are probably arguments to be made for financing its active conservation as a global good. This would require a mechanism over and above the simple accumulation of carbon that comes with increases in vegetative productivity which would already be considered in a global carbon tax mechanism. Land surface albedo modification may pay for itself

with a carbon tax as low at $5/ton (Macias-Fauria et al., 2020), but such measures are probably unfeasible to implement widely in the remaining decades of this century. Since the stewards of the permafrost are by and large relatively poor and marginalized



populations, such a recognition of the monetary value of the stored carbon in the permafrost could serve as a valuable tool for achieving the UN sustainable development goals in the Arctic.

## Code and data availability

All model data used in this work are available from the Earth System Grid Federation (WCRP, 2022; https://esgf-node.llnl.gov/projects/cmip6, last access: 3 July 2022). Standard PInc-PanTher model from Koven et al. (2015); bias correction ISI-MIP method from https://github.com/SantanderMetGroup/downscaleR; PAGE-ICE software from Yumashev et al. (2019).

## Author contributions

YC and AL conceived and designed the analysis. AL collected the data and performed the analysis. AL and YC wrote the
paper, and JCM provided critical suggestions and revised the paper. All authors contributed to the discussion.

## Competing interests

The contact author has declared that neither they nor their co-authors have any competing interests.

## Financial support

This research has been supported by the National Key Research and Development Program of China (grant nos.
2021YFB3900105), the National Natural Science Foundation of China (grant no. 41941006), and Finnish Academy COLD Consortium (grant no. 322430).

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
