# Peer review of "PInc-PanTher estimates of Arctic permafrost soil carbon under the GeoMIP G6solar and G6sulfur experiments"

_Earth System Dynamics, 2022_

## Referee Comment (RC1)

**Peer Review, Liu, et. al., 2022: PInc-PanTher estimates of Arctic permafrost soil carbon under the GeoMIP G6solar and G6sulfur experiments**

The authors use the PInc-PanTher tool to analyze the simulated effects of solar geoengineering (SG) on permafrost carbon. The authors consider six different Earth system models (ESMs) and four different scenarios: moderate emissions (SSP2-4.5), high emissions (SSP5-8.5), high emissions with dimmed sunlight (G6solar) and high emissions with stratospheric aerosol injection, or SAI (G6sulfur). The authors quantify permafrost area, carbon stocks, and economic impacts in each case, and they find that all six ESMs show statistically significant impacts in both SG scenarios.

I thank the authors for the opportunity to review their work. The permafrost-carbon-climate feedback is a critical yet relatively ill-quantified consequence of global warming, and the possible impacts of SG on permafrost carbon are even less well understood. To my knowledge, there have been very few multi-model analyses of the potential effects of SG on permafrost, and this one is well-designed and well-written. My comments are relatively minor, and they largely address word choice, grammar, and clarity. I recommend the manuscript be accepted for publication with minor revisions, and I do not feel it necessary for me to review it again. Specific comments are included in the attached document.

Abstract:
- Line 10: I would be cautious of referring to circumpolar permafrost carbon as a "tipping point"; Schuur, et. al., (2015) argues that while such emissions would be a "significant source" of carbon over decades and centuries, there is little evidence that thawing permafrost would release a "catastrophic" or "large and rapid pulse" of carbon. As such, I would lean towards phrasing like "irreversibility," "positive feedback," and "significant" rather than "tipping point," which implies a sudden or catastrophic change.
- Line 15: There are a lot of numbers in this sentence, which makes it hard to read. Rather than just listing numbers, could you order and phrase them in a way that makes them easier to process? For example, "Under SSP2-4.5 and SSP5-8.5, respectively, permafrost area decreases by ___ and ___ by 2100, and carbon stocks decrease by ___ and ___. In comparison, under G6solar, permafrost area decreases by only ___ and carbon stocks decrease by only ___; under G6sulfur, they decrease by only ___ and ___." Alternatively, you could omit the raw numbers altogether, and write something like

"we find that G6solar mitigates X % of the permafrost area loss for SSP5-8.5 relative to SSP2-4.5".

- Line 18: This sentence is unclear because the subject of the first phrase lacks an object, and it took me several reads to understand the meaning. Either write out the object twice (e.g., "X raises soil C conservation, while Y lowers it, with the net effect of…") or write in the passive voice (e.g., "Soil C conservation is raised by X and lowered by Y, with the net effect of…")

- Line 21: Here and throughout the paper, "ESM" should be used when referring to only one model, and "ESMs" should be used when referring to more than one.

- Line 24: I talk about this more in my comments on the discussion/conclusions below, but I strongly advise cutting the last part of this sentence. This is a purely technical paper and not a social science one; while using the potential economic benefits of SG as income for Indigenous communities is an interesting idea, without any citations, such discussions are pretty far outside the scope of this paper. The abstract especially is much stronger if you stick to the results of the paper, and I'd recommend stopping after "...$20 trillion in economic losses by 2100," which is a very strong result to end on.

Introduction:

- Line 39: Again, I disagree with the characterization of the permafrost-carbon feedback as a "tipping point". I recommend either using different language, or finding other sources to support this characterization. When describing a tipping point, the author cited here (Linton, et. al., 2019) cites his own work (Linton, et. al., 2008), but in that earlier article, Dr. Linton defines a tipping point as "a critical threshold at which a tiny perturbation can qualitatively alter the state or development of a system"; Table 1 of that 2008 article lists the critical threshold of permafrost as "missing", and in Appendix 2, he writes "no studies to date convincingly demonstrate that [permafrost] is a tipping element by our definition". As cited above, more recent research (Schuur et. al., 2015) argues that, while significant and irreversible, the permafrost-carbon feedback will likely scale incrementally over time and with warming, rather than abruptly.

- Lines 51-54: I dislike that a political stance is used in these lines to justify this study's purpose. While the statement "the best option for mitigating climate change is to aggressively cut GHG emissions by switching to clean energy sources" is probably true, and I agree with it, the word "best" is highly unscientific without substantial support. Additionally, the number of other scientists studying SG does not make this study more

or less important. There is plenty of justification for this research based on scientific merit alone, and I recommend cutting these lines entirely and sticking to the argument begun in lines 48-50 and continued in lines 54-59: SG is potentially faster, cheaper, and technically easier than mitigation and $CO_2$ removal, past studies (e.g., Chen, et. al., 2020) find that it might work, and you improve on them here.

- Line 64: Comma between "injection, respectively."
- Line 68: Can you add more detail about the simulations you're analyzing from each ESM? Based on the rest of the study, each ESM appears to have one ensemble member for each of the four scenarios run from 2015-2100, but adding a sentence here (or earlier in this paragraph) would help clarify things, rather than having readers guess based on the plots or go and look up the G6 papers.

Materials and Methods:
- Section 2.1 as a whole: I would consider combining this with section 2.2, and possibly even moving Table 2 to supplementary, as discussed below. After reading the entire paper, it seems to me that the initial C stocks are listed and discussed to explain model differences and provide justification as to why bias correction is necessary, but they are not actually needed for the analysis, as in Lines 149-150, you say that PInc-PanTher only needs TSL, NPP, and RPE, and one of the models also doesn't report initial C stocks. If this is correct, this is good background information and still worth including, but consider consolidating and combining sections 2.1 and 2.2 and moving Table 2 to supplementary. (If this is incorrect, and initial C stocks are used directly by PInc-PanTher, please explain how you can conduct the analysis for the model whose initial C stocks are not reported!)
- Line 83: This sentence needs to be split up, e.g. "CMIP6 ESMs have many improvements, including X, Y, and Z. However, estimates still vary across models, which has been associated with deficiencies in the representation of Q and R."
- Lines 87-88: Not critical, but can you provide any more information on how the other models estimated initial carbon stocks if they didn't use observations? Nothing too detailed, but a few words here would be helpful if you can find them.
- Lines 91-94: "Can be" explained, or "is" explained? According to Table 2, the other models don't underestimate soil C loss; they show no losses at all. I don't understand the cause-and-effect you're trying to argue in the last two sentences in this paragraph; whatever you're trying to say, please try to explain it more clearly.

- Table 2: As mentioned above, consider moving this table to supplementary - the differences between initial carbon stocks are interesting, and mentioning them is important to explain why bias correction is necessary. However, since PInc-PanTher doesn't appear to use them (the numbers are missing for one of the models, and you're still able to conduct your analysis!), they're probably not necessary to include in the main body of the paper.

Results:
- Line 205: The comparison of your results in 2100 to those of McGuire, et. al. (2018) in 2300 raises an eyebrow. You mention that McGuire simulates faster rates of warming, but do the differences between RCP4.5/RCP8.5 and SSP2-4.5/SSP5-8.5 really justify comparing periods 200 years apart? If it does, you should include some more numbers to explain why (what are the rates of warming, exactly?), because otherwise there's a bit of comparing apples to oranges going on here. McGuire finds that, in their study, significant losses of carbon wouldn't occur until after 2100; perhaps digging into that would be a more apt comparison here or in the next subsection.
- Table 3 (and other tables, where applicable): Putting a description of the whole table - "Cooled TSL (℃)" - as the header of the first column is confusing. That information is already in the table description, so I recommend writing "scenario" or similar as the header of the first column.
- Figure 5: What defines the boundaries of the "permafrost region" in this figure? In the paper as a whole, the term appears to refer generally to the area at high latitudes where permafrost can be found, but in this figure, every panel seems to have exactly the same region outlined in color. More specifically, does the red color indicate regions that qualified as permafrost in 2015, or some other period, according to PInc-PanTher, but are no longer permafrost as of 2100?
- Line 237: It's a bit picky, but I'm not sure I would qualify 81 ± 8 vs. 92 ± 17 as "close"... maybe "similar to," "the same order of magnitude" or "within the bounds of uncertainty"?
- Line 249: "the spatial distribution is controlled by latitude and initial C stocks" - I thought all of the PInc-PanTher simulations used identical initial C stocks? This is implied by Lines 149-150, which says that only TSL, NPP, and RPE from each ESM are used; by Lines 154-155, which describes initial C stocks for PInc-PanTher; and by the fact that initial C stocks for one of the models are missing. If this is correct, then consider revising

this statement slightly, perhaps saying that distribution is related to initial C stocks and controlled by the initial distributions of the variables actually fed into PInc-PanTher.

● Section 3.3 in general: Lee, et. al. (2022, https://doi.org/10.1002/essoar.10512047.1) examined high-latitude stratospheric aerosol injection and its effect on permafrost in CESM2-WACCM. That study only used one climate model, but they also looked at the effects on soil carbon vs. vegetation carbon; it may be worth comparing your results here, or in the first or third paragraph of the conclusions.

Discussion and Conclusions:

● Lines 344-345: Can you qualify this statement from McGuire a bit more? Gains in vegetation C might outweigh losses in soil C in the near future, but is that true for any scenario, over any period of time?

● Line 359: Need an "and" after the comma

● Lines 360-end: I'm extremely nervous about telling historically marginalized communities what they "should" do, especially in a purely technical paper. You make some good points here, but as this is an earth science paper and not a social science one, I would strongly recommend leaving most of this implied, but unsaid - you've laid out the evidence for the economic benefits of SG; best leave it up to the social scientists to ponder what to do with them. I would cut this entire block, but if you do want to leave part of it in, I advise sticking to objective sentences that would be very difficult to disagree with.

References:

● Line 467: There's a typo in the DOI for McGuire, et. al. (2018): the DOI has a space before the last two numbers, which makes it a bit harder to follow the link.

---

## Author Comment (AC1)

**I. Response to comments and suggestions of Reviewer #1**

**Peer Review, Liu, et. al., 2022: PInc-PanTher estimates of Arctic permafrost soil carbon under the GeoMIP G6solar and G6sulfur experiments**

The authors use the PInc-PanTher tool to analyze the simulated effects of solar geoengineering (SG) on permafrost carbon. The authors consider six different Earth system models (ESMs) and four different scenarios: moderate emissions (SSP2-4.5), high emissions (SSP5-8.5), high emissions with dimmed sunlight (G6solar) and high emissions with stratospheric aerosol injection, or SAI (G6sulfur). The authors quantify permafrost area, carbon stocks, and economic impacts in each case, and they find that all six ESMs show statistically significant impacts in both SG scenarios.

I thank the authors for the opportunity to review their work. The permafrost-carbon-climate feedback is a critical yet relatively ill-quantified consequence of global warming, and the possible impacts of SG on permafrost carbon are even less well understood. To my knowledge, there have been very few multi-model analyses of the potential effects of SG on permafrost, and this one is well-designed and well-written. My comments are relatively minor, and they largely address word choice, grammar, and clarity. I recommend the manuscript be accepted for publication with minor revisions, and I do not feel it necessary for me to review it again. Specific comments are included in the attached document.

We would like to thank the Anonymous Referee #1 for the appreciation of the main advances of our work. We would also like to thank for all the constructive comments and valuable suggestions, which have helped us to improve the quality of the manuscript. For each question and comment, we gave point-by-point response and made additions and revisions to the manuscript. Please see the attached response.

Abstract:

- Line 10: I would be cautious of referring to circumpolar permafrost carbon as a "tipping point"; Schuur, et. al., (2015) argues that while such emissions would be a "significant source" of carbon over decades and centuries, there is little evidence that thawing permafrost would release a "catastrophic" or "large and rapid pulse" of carbon. As such, I would lean towards phrasing like "irreversibility," "positive feedback," and "significant" rather than "tipping point," which implies a sudden or catastrophic change.

- We have modified the text: "Circum-Arctic permafrost stores large amounts of frozen carbon that must be maintained to avoid catastrophic climate change."

- Line 15: There are a lot of numbers in this sentence, which makes it hard to read. Rather than just listing numbers, could you order and phrase them in a way that makes them easier to process? For example, "Under SSP2-4.5 and SSP5-8.5, respectively, permafrost area decreases by ____ and ____ by 2100, and carbon stocks decrease by ___ and ___. In comparison, under G6solar, permafrost area decreases by only ___ and carbon stocks decrease by only ___; under G6sulfur, they decrease by only ____ and ___." Alternatively, you could omit the raw numbers altogether, and write something like "we find that G6solar mitigates X % of the permafrost area loss for SSP5-8.5 relative to SSP2-4.5".

- We have made changes as suggested. "By 2100, simulations indicate a loss of 9.2 ± 0.4 (mean ± standard error) million km$^2$ of permafrost area and 81 ± 8 Pg of soil carbon under the SSP5-8.5 scenario. In comparison, under SSP2-4.5, G6solar and G6sulfur, permafrost area loss would be mitigated by approximately 39%, 37% and 34% and soil carbon loss by 42%, 54% and 47%, respectively, relative to SSP5-8.5."

- Line 18: This sentence is unclear because the subject of the first phrase lacks an object, and it took me several reads to understand the meaning. Either write out the object twice (e.g., "X raises soil C conservation, while Y lowers it, with the net effect of…") or write in the passive voice (e.g., "Soil C conservation is raised by X and lowered by Y, with the net effect of…")

- We have made changes as suggested. "Increased carbon flux from vegetation to soil raises soil C storage, while the priming effects of root exudates lowers it, with a net mitigating effect on

soil C loss."

- Line 21: Here and throughout the paper, "ESM" should be used when referring to only one model, and "ESMs" should be used when referring to more than one.
- Thanks for reminding. We have made changes as suggested.

- Line 24: I talk about this more in my comments on the discussion/conclusions below, but I strongly advise cutting the last part of this sentence. This is a purely technical paper and not a social science one; while using the potential economic benefits of SG as income for Indigenous communities is an interesting idea, without any citations, such discussions are pretty far outside the scope of this paper. The abstract especially is much stronger if you stick to the results of the paper, and I'd recommend stopping after "...$20 trillion in economic losses by 2100," which is a very strong result to end on.
- We have made changes as suggested.

Introduction:
- Line 39: Again, I disagree with the characterization of the permafrost-carbon feedback as a "tipping point". I recommend either using different language, or finding other sources to support this characterization. When describing a tipping point, the author cited here (Linton, et. al., 2019) cites his own work (Linton, et. al., 2008), but in that earlier article, Dr. Linton defines a tipping point as "a critical threshold at which a tiny perturbation can qualitatively alter the state or development of a system"; Table 1 of that 2008 article lists the critical threshold of permafrost as "missing", and in Appendix 2, he writes "no studies to date convincingly demonstrate that [permafrost] is a tipping element by our definition". As cited above, more recent research (Schuur et. al., 2015) argues that, while significant and irreversible, the permafrost-carbon feedback will likely scale incrementally over time and with warming, rather than abruptly.
- Agreed. We have replaced the reference to Lenton et al., 2019 with the McKay et al., 2022 paper on tipping points that defines abrupt loss of boreal permafrost as a potential tipping point, but with lower confidence than the gradual decline the referee notes:

  "The large hysteresis in the water/ice phase change, and associated climate feedbacks make this an essentially irreversible change, and a potential "tipping point" (McKay et al., 2022)."

  McKay, D.I.A., Staal, A., Abrams J.F., Winkelmann, R., Sakschewski, B., Loriani,S., Fetzer,I.,

Cornell,S.E., Rockström,J., Lenton, T.M.: Exceeding 1.5°C global warming could trigger multiple climate tipping points, Science, https://doi.org/10.1126/science.abn7950, 2022.

- Lines 51-54: I dislike that a political stance is used in these lines to justify this study's purpose. While the statement "the best option for mitigating climate change is to aggressively cut GHG emissions by switching to clean energy sources" is probably true, and I agree with it, the word "best" is highly unscientific without substantial support. Additionally, the number of other scientists studying SG does not make this study more r less important. There is plenty of justification for this research based on scientific merit alone, and I recommend cutting these lines entirely and sticking to the argument begun in lines 48-50 and continued in lines 54-59: SG is potentially faster, cheaper, and technically easier than mitigation and $CO_2$ removal, past studies (e.g., Chen, et. al., 2020) find that it might work, and you improve on them here.

- We have removed them as suggested.

- Line 64: Comma between "injection, respectively."
- Thanks for the correction.

- Line 68: Can you add more detail about the simulations you're analyzing from each ESM? Based on the rest of the study, each ESM appears to have one ensemble member for each of the four scenarios run from 2015-2100, but adding a sentence here (or earlier in this paragraph) would help clarify things, rather than having readers guess based on the plots or go and look up the G6 papers.

- We have added the description. "These six ESMs contain simulations for the G6solar, G6sulfur, SSP5-8.5 and SSP2-4.5 scenarios up to 2100, and their first ensemble members were used for analysis."

Materials and Methods:
- Section 2.1 as a whole: I would consider combining this with section 2.2, and possibly even moving Table 2 to supplementary, as discussed below. After reading the entire paper, it seems to me that the initial C stocks are listed and discussed to explain model differences and provide justification as to why bias correction is necessary, but they are not actually needed for the

analysis, as in Lines 149-150, you say that PInc-PanTher only needs TSL, NPP, and RPE, and one of the models also doesn't report initial C stocks. If this is correct, this is good background information and still worth including, but consider consolidating and combining sections 2.1 and 2.2 and moving Table 2 to supplementary. (If this is incorrect, and initial C stocks are used directly by PInc-PanTher, please explain how you can conduct the analysis for the model whose initial C stocks are not reported!)

- Thanks. We have modified it as suggested.

- Line 83: This sentence needs to be split up, e.g. "CMIP6 ESMs have many improvements, including X, Y, and Z. However, estimates still vary across models, which has been associated with deficiencies in the representation of Q and R."

- We have revised it as suggested. "CMIP6 ESMs have many improvements over preceding generations of CMIP models, including better treatment of snow radiative transfer and insulation effects, soil hydrology and vegetation dynamics (Fox-Kemper et al., 2021). However, estimates of permafrost extent and carbon stock changes still vary across models, which has been associated with deficiencies in the representation of soil thermodynamics and carbon dynamics (Burke et al., 2020; Mudryk et al., 2020)."

- Lines 87-88: Not critical, but can you provide any more information on how the other models estimated initial carbon stocks if they didn't use observations? Nothing too detailed, but a few words here would be helpful if you can find them.

- We have added the description. "Of the six CMIP6 models we used (Table 1), only the CESM2-WACCM land surface model (CLM5) adjusted permafrost carbon stocks to the latest observations and included a vertically resolved soil carbon representation, which is important for more consistent modelling of real-world soil carbon (Varney et al., 2022)."
  Varney, R. M., Chadburn, S. E., Burke, E. J., and Cox, P. M.: Evaluation of soil carbon simulation in CMIP6 Earth System Models, Biogeosciences, https://doi.org/10.5194/bg-19-4671-2022, 2022.

- Lines 91-94: "Can be" explained, or "is" explained? According to Table 2, the other models don't underestimate soil C loss; they show no losses at all. I don't understand the cause-and-effect

you're trying to argue in the last two sentences in this paragraph; whatever you're trying to say, please try to explain it more clearly.

- We have revised it as suggested. "This is explained by the underestimation of initial permafrost CSoil which then leads to an underestimation of future soil C decomposition. CSoil flux from surface vegetation increases due to GHG-driven increases in productivity, and this input flux exceeds soil C decomposition in those models with little initial soil carbon."

- Table 2: As mentioned above, consider moving this table to supplementary - the differences between initial carbon stocks are interesting, and mentioning them is important to explain why bias correction is necessary. However, since PInc-PanTher doesn't appear to use them (the numbers are missing for one of the models, and you're still able to conduct your analysis!), they're probably not necessary to include in the main body of the paper.

- We have moved this table to the supplementary section as suggested.

Results:
- Line 205: The comparison of your results in 2100 to those of McGuire, et. al. (2018) in 2300 raises an eyebrow. You mention that McGuire simulates faster rates of warming, but do the differences between RCP4.5/RCP8.5 and SSP2-4.5/SSP5-8.5 really justify comparing periods 200 years apart? If it does, you should include some more numbers to explain why (what are the rates of warming, exactly?), because otherwise there's a bit of comparing apples to oranges going on here. McGuire finds that, in their study, significant losses of carbon wouldn't occur until after 2100; perhaps digging into that would be a more apt comparison here or in the next subsection.

- Agreed, we have removed the comparison here as suggested, leaving it for the Discussion.

- Table 3 (and other tables, where applicable): Putting a description of the whole table - "Cooled TSL ($^{\circ}$C)" - as the header of the first column is confusing. That information is already in the table description, so I recommend writing "scenario" or similar as the header of the first column.

- We have modified Table 3.

- Figure 5: What defines the boundaries of the "permafrost region" in this figure? In the paper as

a whole, the term appears to refer generally to the area at high latitudes where permafrost can be found, but in this figure, every panel seems to have exactly the same region outlined in color. More specifically, does the red color indicate regions that qualified as permafrost in 2015, or some other period, according to PInc-PanTher, but are no longer permafrost as of 2100?

- We have added the description to the caption. "The initial permafrost extent was derived from the boundary of the permafrost soil C maps (Hugelius et al., 2014)."

- Line 237: It's a bit picky, but I'm not sure I would qualify 81 ± 8 vs. 92 ± 17 as "close"... maybe "similar to," "the same order of magnitude" or "within the bounds of uncertainty"?

- We have revised it as suggested. "Our estimated permafrost C losses for the SSP5-8.5 projection are within the uncertainty range of the 92±17 Pg C estimate based on a literature compilation"

- Line 249: "the spatial distribution is controlled by latitude and initial C stocks" - I thought all of the PInc-PanTher simulations used identical initial C stocks? This is implied by Lines 149-150, which says that only TSL, NPP, and RPE from each ESM are used; by Lines 154-155, which describes initial C stocks for PInc-PanTher; and by the fact that initial C stocks for one of the models are missing. If this is correct, then consider revising this statement slightly, perhaps saying that distribution is related to initial C stocks and controlled by the initial distributions of the variables actually fed into PInc-PanTher.

- Thanks, we have revised it as suggested. "the spatial distribution is related to initial C stocks and controlled by the distributions of TSL, NPP, and RPE."

- Section 3.3 in general: Lee, et. al. (2022, https://doi.org/10.1002/essoar.10512047.1) examined high-latitude stratospheric aerosol injection and its effect on permafrost in CESM2-WACCM. That study only used one climate model, but they also looked at the effects on soil carbon vs. vegetation carbon; it may be worth comparing your results here, or in the first or third paragraph of the conclusions.

- As suggested, we have added citation and comparison. "Lee et al. (2022) studied two Arctic-only stratospheric aerosol injection strategies using CESM2-WACCM and found that the reduction in vegetation carbon gains due to SG outweighed the increase in permafrost soil carbon."

Discussion and Conclusions:

- Lines 344-345: Can you qualify this statement from McGuire a bit more? Gains in vegetation C might outweigh losses in soil C in the near future, but is that true for any scenario, over any period of time?

- We have revised it as suggested. "Gains in vegetation C are expected to partially, or even over-compensate, for losses in soil C in the permafrost region over the century (McGuire et al., 2018)."

- Line 359: Need an "and" after the comma
- Thanks. Corrected.

- Lines 360-end: I'm extremely nervous about telling historically marginalized communities what they "should" do, especially in a purely technical paper. You make some good points here, but as this is an earth science paper and not a social science one, I would strongly recommend leaving most of this implied, but unsaid - you've laid out the evidence for the economic benefits of SG; best leave it up to the social scientists to ponder what to do with them. I would cut this entire block, but if you do want to leave part of it in, I advise sticking to objective sentences that would be very difficult to disagree with.
- We have removed this sentence as suggested.

References:

- Line 467: There's a typo in the DOI for McGuire, et. al. (2018): the DOI has a space before the last two numbers, which makes it a bit harder to follow the link.

- Checked, the DOI can be linked correctly.

---

## Author Comment (AC2)

**III. Response to comments and suggestions of Reviewer #3**

**Peer Review, Liu, et. al., 2022: PInc-PanTher estimates of Arctic permafrost soil carbon under the GeoMIP G6solar and G6sulfur experiments**

This is a really interesting study. The authors have done a careful job and obtained some interesting results. I think that most of this is well done, but I am recommending some revisions.

We would like to thank the Anonymous Referee #3 for the appreciation of the main advances of our work. For each question and comment, we gave point-by-point response and made additions and revisions to the manuscript. Please see the attached response.

My main issue is the econometrics section where you're computing socioeconomic benefits. I'm fine with what you've done, but I think you need to be more careful in your descriptions. There may be socioeconomic harms (or other unforeseen benefits) that you're not discussing because those are not captured in your model. Statements in your abstract like "averting about $20 trillion in economic losses" does not communicate this uncertainty and conveys way too much confidence. There are other examples in the paper that need similar attention.

Thanks for pointing it out. We have added the uncertainty range to statements. For example, in abstract "G6 experiments mitigate ~1/3 of permafrost area loss and halve carbon loss for SSP5-8.5, averting $0–70 (mean 20) trillion in economic losses through reduced permafrost emissions."

Other similar expressions in the manuscript have likewise been modified.

Relatedly, your 90% confidence intervals for economic benefits are approximately $0-70 trillion. Does that mean there is no possibility of harm (negative values)? That requires justification.

The economic impact of SG in reducing GHG emissions from permafrost is negative or greater than 70 trillion in 10% of the 100,000 Monte Carlo simulations. Thus, the harm is present in about 5% of the simulations.

We have added additional notes "In about 5% of the simulations, small negative benefits – that is harm – are predicted, the 2.5 percentiles being $-9 and $-11 trillion for G6solar and G6sulfur, respectively and, for comparison, the corresponding 97.5 percentiles are $90 and $77 trillion."

Figures 1-3: It's hard to see differences between the top and bottom rows. Can you add a third row showing the differences?

Thanks for the suggestion. We have modified the colors in Figures 1-3 to make the scenario differences easier to identify and added Table 2 to show the offsets.

We decided not to add a third row because the trends before and after the bias correction are consistent by design, and the offsets are small.

Table 2. Bias correction mean offsets of TSL, NPP and RPE. Mean offsets of soil temperature (TSL, in °C), net primary productivity (NPP, in g m-2 yr-1), and rhizosphere priming effect (RPE, unitless) for all scenarios over the period 2015–2020. Negative values represent that reference data are smaller than the original ESM simulations.

| Variables | CESM2-WACCM | CNRM-ESM2-1 | IPSL-CM6A-LR | MPI-ESM1-2-LR | MPI-ESM1-2-LR | UKESM1-0-LL |
|---|---|---|---|---|---|---|
| TSL | 1.3 | 1.6 | -1.7 | 3.4 | 4.0 | -0.9 |
| NPP | 10.3 | -88.0 | 69.9 | -37.7 | -51.2 | -31.9 |
| RPE | -0.04 | 0.00 | -0.06 | 0.00 | -0.04 | -0.06 |

Lines 360-361: Stating that Indigenous people should consider geoengineering "with urgency" when they're not the ones capable of deploying geoengineering smacks of colonialism.

Agreed. We certainly did not intend it that way, quite the opposite, in fact. At the suggestion of Reviewer #1, we have decided to remove the statement relating to indigenous peoples from the abstract and discussion.

---

## Author Comment (AC3)

**II. Response to comments and suggestions of Reviewer #2**

**Peer Review, Liu, et. al., 2022: Plnc-PanTher estimates of Arctic permafrost soil carbon under the GeoMIP G6solar and G6sulfur experiments**

Overall this is a good contribution, and most of my comments are with regards to the clarity of presentation (including the current version leaving out a few details that are important). Some of this is simply that as written it implicitly is stating the results of this study as if they were statements about solar geoengineering more generally, vs statements about this particular strategy (tropical injection) in this particular scenario (cooling back down only to SSP2-4.5 levels, so that the global mean temperature continues to increase, just more slowly). Relevant to that it might be useful to try and make some statements in the conclusions about what one might expect to see in other cases, e.g., if SRM were used to hold global mean temperature constant, what would happen; if injection was done at higher latitudes… obviously you can't actually say that without having looked at those simulations, but you could potentially comment at least enough to make it clear that the answers will ultimately depend on the implementation.

We would like to thank the Anonymous Referee #2 for all the constructive comments and valuable suggestions on the previous version of the manuscript, which have helped us to improve the quality of the manuscript. We have brought in more discussion on the more polar-targeted injection schemes that have bene published and emphasized the difference with the G6 tropical injections. For each question and comment, we gave point-by-point response and made additions and revisions to the manuscript. Please see the attached response.

1. L11, SG could slow, could also stop it if one wanted, could even reverse it if one wanted. Why implicitly exclude these other options? (This is written as a generic statement, not a statement about the specific simulations you conducted.)

- Agreed, we have revised it. "Solar geoengineering (SG) has the potential to cool the Arctic surface by increasing planetary albedo"

2. L14-15, I know what you mean but someone unfamiliar with G6 might not realize that the SG is *only* applied for the SSP585. Nor might it be clear to a reader that in these scenarios, SG is used not to stop warming but only to reduce it to SSP2-4.5 levels – this is important context for the conclusions! (Given that G6 appears to roughly restore permafrost conditions to SSP2-4.5 levels also, one might reasonably infer that had more SG been used, one could prevent any further permafrost loss should one choose to.) Nor would a non-SG reader know that G6solar is a solar reduction and G6sulfur is stratospheric sulfate aerosols. (The abstract should be interpretable by people who are not already intimately familiar with GeoMIP scenarios.)

- We have modified the abstract as suggested. "Six earth system models are used to drive the model, running G6solar (solar dimming) and G6sulfur (stratospheric sulfate aerosols) experiments which reduce radiative forcing from SSP5-8.5 (no mitigation) to SSP2-4.5 (substantive mitigation) levels."

3. L15-18, I don't think these numbers are useful to anyone who doesn't already know what G6 scenarios are (per above comment). Might be more useful to say that sufficient SG to yield global mean temperatures consistent with the SSP2-4.5 pathway under SSP2-8.5 $CO_2$ concentrations also leads to permafrost area and soil carbon not statistically significantly different from those under SSP2-4.5, either under solar dimming or stratospheric aerosols.

- We have revised it. "By 2100, simulations indicate a loss of 9.2 ± 0.4 (mean ± standard error) million $km^2$ of permafrost area and 81 ± 8 Pg of soil carbon under the SSP5-8.5 scenario. In comparison, under SSP2-4.5, G6solar and G6sulfur, permafrost area loss would be mitigated by approximately 39%, 37% and 34% and soil carbon loss by 42%, 54% and 47%, respectively, relative to SSP5-8.5."

4. L21-23, (i) The first part of this conclusion is not correct as written, because it is written as if it is a generically true statement about SG rather than a statement about this particular scenario. A reader might reasonably infer that SG actually couldn't do more than this, because that is what the sentence as written (making it a generic statement) implicitly says. I would assume from your results that SG could mitigate all the area loss and carbon loss if we wanted. (ii) the last part of the sentence about an income stream for the Arctic population doesn't seem like a scientific statement but a guess. This isn't an economics or IR paper. (Personally I don't have a problem speculating on this in the conclusions, but highlighting that level of speculation in the abstract of what is otherwise a scientific paper feels a bridge too far.)

- Thanks for pointing this out. We have revised the statement and deleted the last half of the sentence. "G6 experiments mitigate ~1/3 of permafrost area loss and halve carbon loss for SSP5-8.5, averting $0–70 (mean 20) trillion in economic losses through reduced permafrost emissions."

5. L29, missing close quotation. (Plus, there's been fair criticism of continuing to call 5-85 as BAU given the existence of policy changes and pledges; labeling it BAU is inconsistent with the first line of the intro.)

- We have revised it as suggested. "lowering global temperatures from a no-mitigation baseline scenario to a moderate emissions level"

6. L48 is written as if these are alternatives; the point about speed is appropriate but wording could be improved to avoid framing as an either/or. Ditto L51.

- We have revised it and removed policy-relevant statements. "The principal advantage of SG compared with CO2 removal and substantial emission reductions is that temperatures can be reduced far faster; SG may also face fewer technical and financial hurdles (Aldy et al., 2021), however, the potential for damage by SG has not yet been fully explored (Zarnetske et al., 2021). In an earlier study, Chen et al. (2020)…"

7. L55… I think it would be worth defining GeoMIP somewhere in the definition of G4. I guess you

do in the next paragraph…

● We have moved the definition of GeoMIP to the front as suggested. "Chen et al. (2020) found that five of seven CMIP5 Earth System Models (ESMs) driven by the Geoengineering Model Intercomparison Project (GeoMIP) G4 stratospheric aerosol injection geoengineering scheme simulated significant mitigation of Arctic permafrost soil carbon loss."

8. L63… was the target of G6 the radiative forcing, or global mean temperature? (I honestly forget, and I'm on an airplane and not bothering to pay for wifi, so can't look it up.)

● We have checked and it is radiation forcing.

9. L64, repeated "more". But more to the point, this would be a great opportunity to comment on the obvious scenario dependence (including not just the amount of cooling, but things like latitudes of injection). Ultimately (future paper of course) would be good to look at some of the more recent simulations still…

● Thanks for pointing this out. We have added comment "More sophisticated SG deployment strategies are being explored such as the latitudes of injection and its seasonality (Lee et al., 2021, 2022), but are still at the single ESM simulation stage, while the G6 experiments have been simulated by six ESMs (Table 1)."

Lee, W. R., MacMartin, D. G., Visioni, D., Kravitz, B.: High-latitude stratospheric aerosol geoengineering can be more effective if injection is limited to spring. Geophys. Res. Lett., https://doi.org/10.1029/2021GL092696, 2021

Lee, W. R., MacMartin, D. G., Visioni, D., Kravitz, B., Chen, Y., Moore, J. C., Leguy, G., Lawrence, D. M., and Bailey, D. A.: High-latitude stratospheric aerosol injection to preserve the Arctic., Earth and Space Science Open Archive [preprint], https://doi.org/10.1002/essoar.10512047.1, 22 November 2022

10. L110, should define TSL, NPP, and GPP. (Even if I know what they are… other readers might not)

- We have explained TSL, NPP and GPP where they first appear (the caption of Table 1).

11. Figures 1-3, when I can't see the G6solar line, is it under G6sulfur?

- Yes. We have modified the colors in Figures 1-3.

12. L170, why is 2015-19 in equilibrium?

- We have added explanations. "The initial carbon flux into the soil pool is inferred from the initial steady state, which satisfies the condition that soil C loss and input are in equilibrium during the first five years (2015–2019). This initial equilibrium assumption ignores decomposition occurring in the active layer in the current climate and aims to remove the effects of decomposition that would also occur under a constant climate for predicting the response of soil carbon loss to future soil warming (Koven et al., 2015)."

13. Section 2.3 more generally… there are certainly some assumptions that go into this model; it might be useful somewhere to give some indication for which ones importantly affect results and which don't, and how significantly they affect things. (E.g., if the 2.5-fold increase for 10C change was 2.0, or 3.0, would that radically change conclusions?  Is that sort of uncertainty likely?)

- This is an interesting question, but beyond the scope of this study. However, rates of chemical reactions generally double for temperature rises of about 10C but can be somewhat faster in living tissue. Hence, it would be very unlikely for the three- or five-times higher activation energies, suggested as examples by the referee to double reaction rates in 2 or 3C, to occur in decaying tissues. We have added "The parameters of PInc-PanTher model are derived from laboratory incubation syntheses and literature reviews. For a more detailed description, please refer to Koven et al. (2015)."

14. Section 2.4, there will be some rather critical assumptions in here too, which aren't even stated. Like ratio of C emitted as CO2 vs CH4.  Or the discount rate.  Again, it's ok to refer to published

literature, but giving some context (to save us from looking things up) would be useful, and to the extent possible worth acknowledging degree of uncertainty.

- Thanks for this point. We have clarified the proportion of $CH_4$ emissions in section 3.4. "We assume that $CH_4$ emissions are 2.3% of the overall soil respiration rate based on available studies and expert assessments (Schuur et al., 2013; Schneider Von Deimling et al., 2015; Gasser et al., 2018), and that the remaining soil carbon is emitted as $CO_2$."

  And we have added notes on the PAGE-ICE parameters. "Finally, the estimated climate …, converting changes in consumption to utility through the elasticity of marginal utility (EMUC) to correct for regional income differences, and discounting aggregates based on the pure time preference (PTP) rate. In PAGE-ICE, both PTP and EMUC follow triangular distributions ranging from 1 (0.1–2) and 1 (0.5–2) respectively."

15. L205-6, minor quibble, but could you put the RCP8.5 and 4.5 in the same order as in the previous sentence?

- We have removed the comparison with McGuire et al. (2018) here, as suggested by reviewer #1, because the time span of the study is so different.

16. L211-214, not sure the G6solar vs sulfur results are even statistically significant, but worth saying more here. When people ran the simulations to achieve SSP2-4.5 temperatures (or RF), were the global mean values for G6solar the same as G6sulfur? (That is, some effect could simply be how well they executed the G6 protocol.) Or, if the modelers perfectly balanced RF in each case, did that also manage temperature equally well in both cases? Second, for same global mean temperature under the two, is the typical overcooling of tropics / undercooling of high latitudes the same for the solar and sulfur simulations? (Given that AOD is likely higher in the tropics for the specific G6 protocol, I might expect more tropical overcooling than for G6solar, leading to a physical reason why G6sulfur as specified might be worse than G6solar for permafrost, but that would be a result of the G6 specification, not a feature inherent to SAI vs solar reduction, indeed SAI would presumably give more flexibility to alter latitudinal dependence.) Or, is any difference between G6solar and G6sulfur due to something associated with the aerosols themselves somehow? (E.g., assumptions in the land model and

how it handles direct to diffuse light.) I think it would be both easy and important to check the first two possible sources of difference. It would also be worth pointing out somewhere that G6sulfur assumes tropical injection, which tends to undercool high latitudes relative to low, and that that is a choice; that other choices for injection latitude might do relatively more cooling at higher latitudes.

- Visioni et al., 2021 report detailed analysis of the surface climate response for the 6 ESMs running the G6 experiments. For the purposes of this answer, we focus on surface temperatures only, while remembering that other elements of the system also are affected. From Visioni et al., 2021: "All models successfully reduce global mean surface air temperatures to within 0.2 ∘C of SSP2-4.5 levels on average throughout the century with both geoengineering methods, but the amount of geoengineering required to do so varies across models."

   So, we can answer your first question: "how well they executed the G6 protocol / manage temperature equally well in both cases?" We conclude that all models performed the G6 experiments adequately since residual surface temperature differences G6-SSP245 are significantly larger than 0.2 ∘C of SSP2-4.5 in various locations.

   The second question: "for same global mean temperature under the two, is the typical overcooling of tropics / undercooling of high latitudes the same for the solar and sulfur simulations?"

   No, it is different. G6sulfur has more residual temperature differences than G6solar just as the referee expects. There is also far more across-ESM variability under G6sulfur than G6solar (see fig 7 in Visioni et al., 2021).

   The third question: "is any difference between G6solar and G6sulfur due to something associated with the aerosols themselves somehow?"

   The response in the Arctic under G6sulfur is clearly due to the aerosols. Probably due to stratospheric heating (Visioni et al., 2021 and references therein). The differences between ESM may well be due to different initial states of long-period variations in the ocean or sea ice, but changes in the land vegetation have not been explored to date. As the referee

suggests, the fact that G6 specifies tropical injection, and also specifies the target of global mean temperatures is very important in producing the strongly undercooled polar regions.

We added text: "It is perhaps surprising that the simulated permafrost area loss (and carbon loss, see next section) are so similar under both G6 scenarios, given that G6sulfur has more residual temperature differences and across-ESM variability than G6solar (Visioni et al., 2021). Under both G6 experiments, but especially G6sulfur, the polar regions are undercooled as is the boreal permafrost zone. This is not inherent to stratospheric aerosol injection, but a consequence of the G6 specification of tropical injection and global radiative forcing (hence temperature) target. The bias correction procedure for TSL removes much of the across-ESM differences and offsets from observational data at the start of the simulation period (Fig.1 and Table 2). G6sulfur produces a weaker response with larger uncertainty than G6solar…"

17. L273-274, is that RPE statement for all cases? (It follows a sentence about G6; unclear whether it is intended to be specific to that)

- We have revised the wording. "Specifically, RPE will increase the net loss of soil C, causing additional permafrost emissions of about 6 to 10 Pg C between 2020 and 2100 under the four scenarios".

18. L281… the changed direct and diffuse ratio is only present in G6sulfur yet the sentence talks about both. (Also, relevant to that, do you know how the land models in the various models handles direct to diffuse ratio for driving vegetation?)

- Thanks for pointing it out. We're not quite sure how the land models handle the direct and diffuse radiative forcing, and changes in precipitation under SG will also affect NPP. The expression has been modified. "Increasing input C flux allows more carbon to be retained in the permafrost soils, thus the lower C loss and larger uncertainty under the G6solar and G6sulfur scenarios compared with SSP2-4.5 is mainly caused by the large across model spread in NPP change (Fig. 8)."

19. And Figure 9… wow, that's remarkable! I suppose not really that relevant here, but noting that

CESM G6sulfur does show the "over" cooling in summer as suggested by Jiang et al.

- Thanks. Good to know.

20. L319, are the conclusions very sensitive to this highly-uncertain numbr?

- Methane emissions are presumed to be 1.5%-3.5% of overall soil respiration rates, with the most common value taken as 2.3% (Schuur et al., 2013; Schneider Von Deimling et al., 2015; Gasser et al., 2018). The uncertainty in the methane emission estimates is therefore much smaller than that in the economic impact assessment.

21. L326, whoa… I think you need to say more than just "various". What parameters did you change, and why, and for what range?  Are you coming up with a range of carbon emissions (in which case, shouldn't it be in the previous section)?  Or just a range of economic damages for a given carbon?  (in which case, are you missing the dominant uncertainties?)

- We have revised the wording. "All reported results are derived from an ensemble of 100,000 Monte Carlo simulations to perturb model parameters related to GHG emissions, climate modelling, economic damages and discounting, explore uncertainties in the economic impacts of climate damages, and establish probability distributions of the results."

22. L356, again, the statement here is worded as a generic thing (as if "implementation of SG" was a binary choice, rather than something that one could do more or less of, as well as depending on latitude of injection)

- Agreed, we have revised the wording. "Implementation of G6 experiments on the basis of the SSP5-8.5 scenario could…"

23. L358, but now, for costs, you specify G6 and then say it depends on scenario? This is not well worded… (that is, the "G6 scheme" I think means specifically following G6, i.e., for SSP5-8.5 emissions, with a target of 2-4.5 levels, using tropical injection.  If you meant SAI more generally, you should say that).

- Thanks. We have revised the wording. "The implementation cost of stratospheric aerosol injection is strongly dependent on injection scenario specified, but for high greenhouse gas emission and a consequent degree of cooling, comparable to the G6sulfur scenario, is estimated at $30-70 billion/year (Smith, 2020). Our experiments thus show that SG, at least as defined by G6, but also likely including other schemes that target polar regions more specifically, has considerable global economic benefits even if only the permafrost carbon is included in the calculations."

24. L360… considerable economic benefits even if only the permafrost carbon is included in the calculation!

- Yes!